# Algorithm- and Data-Dependent Generalization Bounds for Diffusion Models

**Benjamin Dupuis**[*1,2], **Dario Shariatian**[*1,2,3], **Maxime Haddouche**[*1,2],
**Alain Durmus**[†3], **Umut Simsekli**[†1,2]

[1] INRIA, France
[2] CNRS, Ecole Normale Supérieure, PSL Research University, France
[3] Ecole Polytechnique, CMAP, IP Paris, France

[*†] indicate equal contributions.

## Abstract

Score-based generative models (SGMs) have emerged as one of the most popular classes of generative models. A substantial body of work now exists on the analysis of SGMs, focusing either on discretization aspects or on their statistical performance. In the latter case, bounds have been derived, under various metrics, between the true data distribution and the distribution induced by the SGM, often demonstrating polynomial convergence rates with respect to the number of training samples. However, these approaches adopt a largely approximation theory viewpoint, which tends to be overly pessimistic and relatively coarse. In particular, they fail to fully explain the empirical success of SGMs or capture the role of the optimization algorithm used in practice to train the score network. To support this observation, we first present simple experiments illustrating the concrete impact of optimization hyperparameters on the generalization ability of the generated distribution. Then, this paper aims to bridge this theoretical gap by providing the first algorithmic- and data-dependent generalization analysis for SGMs. In particular, we establish bounds that explicitly account for the optimization dynamics of the learning algorithm, offering new insights into the generalization behavior of SGMs. Our theoretical findings are supported by empirical results on several datasets.

## 1 Introduction

Score-based Generative Models (SGMs) are among the most popular classes of generative models [HJA20a, SSDK+21, DN21, KAAL22a, EKB+24], with applications ranging from computer vision and medicine to natural language processing; see [YZS+24] for a recent survey.

The starting point of Score-based Generative Models (SGMs) is to consider a stochastic process $(\overrightarrow{X}_t)_{t \in [0,T]}$, referred to as the forward process, which is the solution of an ergodic diffusion over the time interval $[0, T]$ and initialized from the data distribution $\mu$. Typically, $(\overrightarrow{X}_t)_{t \in [0,T]}$ is either a $d$-dimensional Brownian motion or an Ornstein–Uhlenbeck process, with stationary distribution given by the standard Gaussian, denoted by $\gamma^d$ [SSDK+21]. In this work, we focus on the latter case. This construction defines a path measure connecting $\mu$ to $\gamma^d$, and thus an ideal generative model can be formed, by taking a large value of $T$ and considering the time-reversed process associated with $(\overrightarrow{X}_t)_{t \in [0,T]}$, defined for any $t \in [0, T]$ as $\overleftarrow{X}_t = \overrightarrow{X}_{T-t}$. It turns out that the backward process is itself a diffusion process, whose (non-homogeneous) drift $(t, x) \mapsto s(t, x)$ depends on the Stein scores of the forward marginals [HP86], which can be characterized as the solution to a regression problem [Vin11, Hyv05] involving simulations of the forward process $(\overrightarrow{X}_t)_{t \in [0,T]}$. This drift can

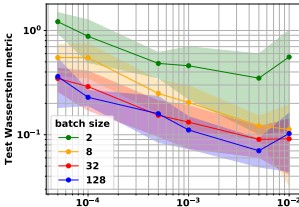 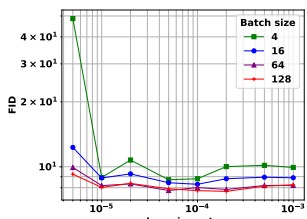 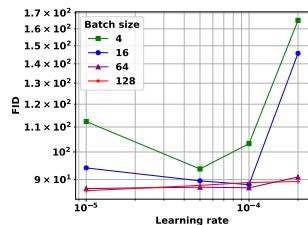

Figure 1: Experiments with varying learning rates and batch sizes obtained with the ADAM optimizer. *(left)* test Wasserstein-2↓ metric on a Gaussian mixture dataset *(middle)* FID↓ on MNIST *(right)* FID↓ on the `butterflies` dataset [WME09]. See Section 4.1 for full experimental details.

be estimated using a family of neural networks $s_\theta : (t, x) \mapsto s_\theta(t, x)$ parameterized by $\theta \in \Theta$ [SDME21]. Once the parameter $\hat{\theta}$ has been learned, an approximation of the backward process can be simulated by starting from $\gamma^d$ and applying a numerical scheme to discretize the corresponding stochastic differential equation, using the approximate score $s_{\hat{\theta}^{(n)}}$ in place of the true score function $s$, where $\hat{\theta}^{(n)}$ is the parameter obtained by the learning procedure and $n$ is the number of data points. Because of their practical relevance, providing performance guarantees for SGMs has received increasing attention in recent years [DBTHD21, LDQ24]. A popular line of research [LLT22b, CCL+23, CLL23, LLT22a] provides theoretical guarantees on the discrepancy between the true data distribution and the generated distribution in various metrics; in particular we focus here on the Kullback Leibler (KL) divergence. More precisely, denoting by $\nu_T^{(n)}$ the distribution of the SGM, the KL divergence of the data distribution with respect to $\nu_T^{(n)}$ can be bounded by three terms which are each associated with one type of approximations of the backward process:

$$\mathrm{KL}(\mu | \nu_T^{(n)}) \lesssim \mathscr{E}_{\mathrm{i}} + \varepsilon_{\mathrm{s}}^{(n)}(\hat{\theta}^{(n)}) + \mathscr{E}_{\mathrm{d}} \,, \tag{1}$$

where (i) $\mathscr{E}_{\mathrm{i}}$ accounts for the fact that the initialization is taken as $\gamma^d$ and not the distribution of $\overrightarrow{X}_T$ (ii) $\varepsilon_{\mathrm{s}}^{(n)}(\hat{\theta}^{(n)})$ accounts for the approximation of $s$ by $s_{\hat{\theta}}$ and (iii) $\mathscr{E}_{\mathrm{d}}$ accounts for the discretization error since in general solving the backward stochastic differential equation (SDE) is not an option even if we would have access to the true score function.

Several studies provide quantitative bounds for the first and last terms $\mathscr{E}_{\mathrm{i}}$ and $\mathscr{E}_{\mathrm{d}}$, which do not depend on the training data and the optimization algorithm that provides $\hat{\theta}^{(n)}$, and they make the underlying assumption that the second term $\varepsilon_{\mathrm{s}}^{(n)}(\hat{\theta}^{(n)})$ is small, *i.e.*, the score network is a good approximation of the true score of the forward process. This makes the bounds completely neglect the impact of the training set and the training algorithm used in practice. This question is at the core of generalization properties of SGMs since training a perfect score model on the empirical dataset would result in straight-up memorization of the dataset, entailing a non-negligible score error with respect to the true data distribution in the finite data regime [LCL24, YSL23].

A popular approach for analyzing the statistical properties of score-based generative models (SGMs) is to rely on approximation theory. The goal is to show that, within a given class of functions $\{s_\theta : \theta \in \Theta\}$, for any number of samples $n \geqslant 1$, there exists a score estimator $s_{\hat{\theta}_\star^{(n)}}$ such that $\varepsilon_{\mathrm{s}}^{(n)}(\hat{\theta}_\star^{(n)}) \leqslant C/n^{\alpha_\mu}$ for $\alpha \in [0, 1]$ which depends on intrinsic properties of $\mu$ and a constant $C$, both being independent of $n$. By combining such results with existing discretization error bounds, one can derive statistical guarantees for SGMs under various metrics. In the continuous-time setting (*i.e.*, $\mathscr{E}_{\mathrm{d}} = 0$), a score approximation rate of order $n^{-\mathcal{O}(\alpha/d)}$ was obtained in [OAS23], where $\alpha$ is a parameter related to the smoothness of the data distribution. As this rate deteriorates exponentially with the ambient dimension $d$, several studies have proposed relying on geometric assumptions on the data distribution, such as the manifold hypothesis [Bor22] (*i.e.*, the support of $\mu$ lies on a bounded submanifold of dimension $d_\mu \leqslant d$). In this context, also using neural networks, it has been shown in [ADR24] that a rate of order $n^{-\mathcal{O}(\alpha/d_\mu)}$ can be achieved, where $\alpha$ again depends on the smoothness of $\mu$. Alternatively, approximation guarantees for neural networks were also established in [CL24] by relying on a notion of complexity of the relative density of $\mu$. Similar approximation results have also been obtained for neural networks in the so-called neural tangent kernel regime [HRX24], as well as for kernel-based score estimators [WWY24, ZYLL24, DKXZ24].

Despite recent advances, this approach suffers from two main limitations: *(i)* the considered class of score estimators is sometimes far from those used in practice (e.g., UNet architectures [HJA20a]), and, more importantly, *(ii)* it does not account for the impact of the learning algorithm (e.g., ADAM,

SGD) used in practice to obtain an estimator $\hat{\theta}^{(n)}$ on the generalization error. Indeed, while existing works establish existence results, the generalization error associated with the actual parameter $\hat{\theta}^{(n)}$ returned by a learning algorithm remains unknown. In this paper, we argue that the learning phase has a significant influence on the error of SGMs. We briefly illustrate this in Figure 1, which shows the effect of ADAM optimizer hyperparameters (learning rate and batch size) on generation performance across three datasets—a Gaussian mixture model, MNIST [LBBH98], and the `butterflies` dataset [WME09]—we observe that hyperparameters clearly influence performance as measured by the Wasserstein distance and Fréchet Inception Distance (FID). Such algorithm-dependent behavior has also been observed in [SOE+24, Figures F.7 and F.8].

Recently, several studies have proposed analyses that aim to account for both the impact of the learning algorithm and the data properties in the study of SGMs. To the best of our knowledge, the first attempt in this direction was made in [LLZB23], which analyzes an idealized learning algorithm consisting of gradient flow in a random feature model in the infinite-width limit. Alternatively, other works have adopted information-theoretic tools to derive generalization bounds, as in [CZS25]. Our work complements these efforts by taking an orthogonal approach and addressing some of their limitations. In particular, the bounds in [CZS25] involve only an implicit dependence on the learning algorithm, making the bounds rather abstract and difficult to assess the actual effect of algorithm dependence on generalization. Moreover, their analysis does not incorporate the training set.

**Contributions.** Relying on an alternative approach, we propose a framework to derive data- and algorithm-dependent generalization bounds for SGMs. Our main contributions are as follows.

- **Generalization adapted decomposition.** In Section 3.1, we provide a key decomposition of $\varepsilon_{\mathrm{s}}(\theta)$ for any $\theta \in \Theta$, informally stated as

$$\varepsilon_{\mathrm{s}}^{(n)}(\theta) = \mathscr{L}_{\mathrm{ESM}}^{(n)}(\theta) + \Delta_{\mathrm{s}}^{(n)} + \mathscr{G}_{\mathrm{l}}^{(n)}(\theta) \,,$$

where $\theta$ is any parameter of the score network. This decomposition highlights three distinct contributions. First, the explicit score matching loss $\mathscr{L}_{\mathrm{ESM}}^{(n)}$ (see Equation (10)) that is optimized during the learning phase. Second, the data-dependent constant $\Delta_{\mathrm{s}}^{(n)}$ is a concentration term capturing the interconnection between the data distribution, the dataset, and the forward process. Finally, $\mathscr{G}_{\mathrm{l}}^{(n)}(\theta)$ is a *score generalization gap*, quantifying the difference between a risk that measures the quality of the score estimation and its empirical counterpart at parameter $\theta$.

- **Characterizing $\Delta_{\mathrm{s}}^{(n)}$ and $\mathscr{G}_{\mathrm{l}}^{(n)}(\theta)$.** We provide quantitative upper bounds on $\Delta_{\mathrm{s}}^{(n)}$ in Section 3.2, and by making connections to smooth Wasserstein distance [NGK21] we show that it is of order $\mathcal{O}(1/\sqrt{n}) + \mathscr{E}_{\mathrm{d}}$ (with a potentially large constant). We then show that $\mathscr{G}_{\mathrm{l}}^{(n)}(\theta)$ is directly amenable to existing learning theoretic tools and we use two existing algorithm- and data-dependent bounds [MWZZ18, ADS+24] that cover a broad range of algorithms. Combined with our theory, these bounds suggest that the gradient norms and the topological properties of optimization trajectories can provide useful information about the generalization performance of SGMs. Furthermore, since $\mathscr{G}_{\mathrm{l}}^{(n)}(\theta)$ is also of order $\mathcal{O}(1/\sqrt{n})$, this ultimately gives us a high-probability bound of the form $\mathscr{L}_{\mathrm{ESM}}^{(n)}(\theta) + \mathcal{O}(1/\sqrt{n}) + \mathscr{E}_{\mathrm{i}} + \mathscr{E}_{\mathrm{d}}$ on the KL divergence between the true data distribution and the generated distribution.

- **Experimental validation.** We design low and high dimensional experiments to validate our theory on different algorithms, varying optimizers (SGLD, ADAM), learning rates and batch sizes. Our make our implementation publicly available in https://github.com/darioShar/Generalization-Diffusion-Models.

**Notation.** For two probability measures $\mu$ and $\nu$, the absolute continuity of $\mu$ with respect to $\nu$ is denoted by $\mu \ll \nu$. The Kullback-Leibler divergence of $\mu$ with respect to $\nu$ is $\mathrm{KL}(\mu|\nu) := \int \log(\mathrm{d}\mu/\mathrm{d}\nu)\mathrm{d}\mu$ if $\mu \ll \nu$, and $\mathrm{KL}(\mu|\nu) := +\infty$ otherwise. We also define the Fisher information of $\mu$ with respect to $\nu$ as $\mathscr{I}(\mu|\nu) := \int \|\nabla \log(\mathrm{d}\mu/\mathrm{d}\nu)\|^2 \mathrm{d}\mu$. We denote by $\gamma^d$ the standard $d$-dimensional Gaussian distribution. For any random variable $Y$, we denote by $\mathrm{Law}(Y)$ its distribution. We denote $\mathrm{B}(0, D)$ the ball of radius $D$, centered at $0$. We write $A \lesssim B$ whenever $A \leqslant CB$ for a universal constant $C$ that neither depends on the assumption's constants nor the parameters at hands.

## 2  Background on Score Generative Models

**Forward and backward process.**   We consider as the forward process a standard $d$-dimensional Ornstein-Uhlenbeck process, solution of the SDE starting from $\mu$:

$$\mathrm{d}\overrightarrow{X}_t = -\overrightarrow{X}_t\mathrm{d}t + \sqrt{2}\mathrm{d}\mathrm{B}_t \ , \quad \overrightarrow{X}_0 \sim \mu \ , \tag{2}$$

where $(\mathrm{B}_t)_{t\geqslant 0}$ is a standard $d$-dimensional Brownian motion. Denote by $\overrightarrow{p}_t$ the density of $\overrightarrow{X}_t$ with respect to the Lebesgue measure and by $\tilde{p}_t := \overrightarrow{p}_t/\gamma^d$ its density with respect to $\gamma^d$, for $t \geqslant 0$. We assume that $\mu$ has a density with respect to the Lebesgue measure, and $\overrightarrow{p}_0$ denotes this density.

Under mild regularity conditions [And82, HP86], the time-reversal $(\overleftarrow{X}_t)_{0\leqslant t\leqslant T}$ of $(\overrightarrow{X}_t)_{0\leqslant t\leqslant T}$ over a time interval $[0,T]$ for some time horizon $T > 0$, defined by $\overleftarrow{X}_t = \overrightarrow{X}_{T-t}$, is solution of the SDE[1]

$$\mathrm{d}\overleftarrow{X}_t = \{-\overleftarrow{X}_t + s(T - t, \overleftarrow{X}_t)\}\mathrm{d}t + \sqrt{2}\mathrm{d}\mathrm{B}_t \ , \quad \overleftarrow{X}_0 \sim \mathrm{Law}(\overrightarrow{X}_T) \ , \tag{3}$$

where we define the score function $s(t, x) = 2\nabla \log \tilde{p}_t(x)$ for any $t \in [0, T]$ and $x \in \mathbb{R}^d$. Note that $(\mathrm{B}_t)_{t\geqslant 0}$ denotes a standard $d$-dimensional Brownian motion, which is distinct from the one used in (2). However, for notational simplicity and by convention, we use the same symbol. The function $(t, x) \mapsto \nabla \log \tilde{p}_t(x)$ is known as the score function. In practice, simulating the backward process is infeasible, and approximations are required. The first challenge arises from the fact that the score function is unknown. However, it can be estimated from data sampled from $\mu$ as described below.

**Score Estimation.**   Using Fisher's identity [Efr11], it is well-known that the score function $(t, x) \mapsto s(t, x)$ satisfies for $t > 0$, $s(t, \overrightarrow{X}_t) = 2\mathbb{E}[\nabla \log \tilde{p}_{t|0}(\overrightarrow{X}_t|\overrightarrow{X}_0)|\overrightarrow{X}_t]$ , where $\tilde{p}_{t|0}$ denotes the conditional density of $\overrightarrow{X}_t$ given $\overrightarrow{X}_0$, with respect to $\gamma^d$. Therefore, it is the solution of a regression problem. Based on a parametric family $\{(t, x) \mapsto s_\theta(t, x) \ : \ \theta \in \Theta\}$, typically neural networks with $\theta$ denoting their weights, we can then learn the parameter $\theta$ by minimizing the *population risk* $\theta \mapsto \mathbb{E}[\ell_\varpi(\theta, Z)]$, associated with denoising loss function, where $Z$ is a sample from $\mu$, $\theta \in \Theta$, $z \in \mathbb{R}^d$ and $\varpi$ a probability distribution over $\mathbb{R}_+$, as:

$$\ell_\varpi(\theta, z) := \int \mathbb{E}[\|s_\theta(t, \overrightarrow{X}_t^z) - 2\nabla \log \tilde{p}_{t|0}(\overrightarrow{X}_t^z|z)\|^2]\mathrm{d}\varpi(t) \ , \tag{4}$$

where $\overrightarrow{X}_t^z$ indicates the forward process (2) with initial value $\overrightarrow{X}_0 = z$. In practice, we need to rely on the empirical risk associated to a dataset $\mathbf{Z}^{(n)} = (Z_1, \ldots, Z_n) \sim \mu^{\otimes n}$ of i.i.d. samples from $\mu$. Therefore, a *learning algorithm* (e.g., SGD or ADAM) is used for obtaining a parameter $\hat{\theta}^{(n)}$ by minimizing the following empirical denoising score matching loss:

$$\mathscr{L}_{\mathrm{DSM}}^{(n,\varpi)}(\theta) := \frac{1}{n}\sum_{i=1}^{n}\ell_\varpi(\theta, Z_i) \ . \tag{5}$$

**Backward simulation.** Once an estimate $\hat{\theta}^{(n)}$ has been obtained, the corresponding score network $s_{\hat{\theta}^{(n)}}$ is used for approximately simulating Equation (3). However, even when replacing the true score in (3) with this estimator, the resulting SDE cannot be solved explicitly. In practice, we must address two main challenges: *(i)* the initial distribution is intractable, and *(ii)* it cannot be exactly simulated. To overcome the first issue, we exploit the fact that the OU process converges geometrically fast to the standard Gaussian distribution $\gamma^d$, and use this as the initialization of our model. For the second issue, we rely on a discretization scheme. In this work, we focus on the exponential integrator (EI) scheme [DM15], which has also been adopted in several recent studies [CDS25, BBDD24, ADR24, ZC23].

Let $N \in \mathbb{N}^\star$ and $h > 0$ be the step size, with $T = Nh$. Define the time steps by $t_k := kh$ for $k \in 1, \ldots, N$. The Euler EI scheme is then defined as follows: starting from $\widehat{X}_0^{(n)} \sim \gamma^d$, for each $k \in 1, \ldots, N$, and given $\widehat{X}_{t_k}^{(n)}$, the trajectory $(\widehat{X}_t^{(n)})_{t\in[t_k, t_{k+1}]}$ is the solution to a linear SDE.

$$\mathrm{d}\widehat{X}_t^{(n)} = (-\widehat{X}_t^{(n)} + s_{\hat{\theta}^{(n)}}(T - t_k, \widehat{X}_{t_k}^{(n)}))\mathrm{d}t + \sqrt{2}\mathrm{d}\mathrm{B}_t \ . \tag{6}$$

We denote by $\nu_t^{(n)}$ the distribution of $\widehat{X}_t^{(n)}$ and refer to it as the *generated* distribution.

**Convergence bounds.** To avoid technicalities and in particular the use of early stopping procedure, we rely for our anlaysis on the following assumption following [CDS25]:

---

[1]Note that we consider the density of $\overrightarrow{X}_t$ with respect to $\gamma^d$, leading to a negative linear drift in (3) [CDS25].

**Assumption 2.1.** *The Fisher information between $\mu$ and $\gamma^d$ is finite, i.e., $\mathscr{I}(\mu|\gamma^d) < \infty$.*

A major challenge emerging from the above procedure is to control the discrepancy between $\mu$ and $\nu_T^{(n)}$. In particular, under Assumption 2.1, [CDS25, Theorem 1] states the following bound.

**Theorem 2.1.** *Under Assumption 2.1, for any $h > 0$ and $N \in \mathbb{N}$ such that $T = hN$, it holds*

$$\mathrm{KL}(\mu|\nu_T^{(n)}) \lesssim \mathrm{e}^{-2T}\mathrm{KL}(\mu|\gamma^d) + T\varepsilon_{\mathrm{s}}^{(n)}(\hat{\theta}^{(n)}) + h\mathscr{I}(\mu|\gamma^d), \tag{7}$$

*where $\varepsilon_{\mathrm{s}}^{(n)}(\theta) := T^{-1}\sum_{k=0}^{N-1} h\mathbb{E}\left[\|s_\theta(T - t_k, \overrightarrow{X}_{T-t_k}) - 2\nabla\log\tilde{p}_{T-t_k}(\overrightarrow{X}_{T-t_k})\|^2\right].$*

The two terms accompanying $\varepsilon_{\mathrm{s}}^{(n)}$ are specific to the approximations involved in modeling the backward process underlying the diffusion model we consider. The first term accounts for the fact that the diffusion model is initialized from $\gamma^d$ rather than from $\mathrm{Law}(\overrightarrow{X}_T)$. The second term corresponds to the discretization error introduced by using the exponential integrator (EI) scheme. Finally, the quantity $\varepsilon_{\mathrm{s}}^{(n)}$ reflects the quality of the score approximation achieved by the score network. We note that several studies have established other guarantees for SGMs under various assumptions, in which $\varepsilon_{\mathrm{s}}^{(n)}$ naturally appears [CDS25, BBDD24, CLL23, CCL+23, LLT22b].

## 3 Towards a better understanding of generative performance

This section proposes a more in-depth study of $\varepsilon_{\mathrm{s}}^{(n)}$. Informally, we show in Section 3.1 the following decomposition $\varepsilon_{\mathrm{s}}^{(n)}(\theta) = \mathscr{L}_{\mathrm{ESM}}^{(n)}(\theta) + \Delta_{\mathrm{s}}^{(n)} + \mathscr{G}_{\mathrm{l}}^{(n)}(\theta)$, where $\mathscr{L}_{\mathrm{ESM}}^{(n)}(\theta)$ is defined in (10), and $\Delta_{\mathrm{s}}^{(n)}, \mathscr{G}_{\mathrm{l}}^{(n)}(\theta)$ highlight respectively the influence of: *(i)* statistical behavior of the training set, *(ii)* the problem of learning $s_\theta$. We then upper-bound $\Delta_{\mathrm{s}}^{(n)}$ in Section 3.2.

### 3.1 A key decomposition

First, we define the following probability measure over $\mathbb{R}_+$:

$$\lambda := \frac{h}{T}\sum_{k=0}^{N-1}\delta_{T-t_k}, \tag{8}$$

where $\delta$ denotes the Dirac measure. For ease of notation, we denote $\mathscr{L}_{\mathrm{DSM}}^{(n,\lambda)}$ by $\mathscr{L}_{\mathrm{DSM}}^{(n)}$; see (5).

With these notations, it is clear that $\mathscr{L}_{\mathrm{DSM}}^{(n)}(\theta) = \widehat{\mathcal{R}}_{\mathbf{Z}^{(n)}}^{(\lambda)}(\theta) := n^{-1}\sum_{i=1}^{n}\ell_\lambda(\theta, Z_i)$, which we refer to as the *empirical risk*, following standard terminology in learning theory. This observation naturally leads to the definition of the corresponding *population risk*, $\mathcal{R}^{(\lambda)}(\theta) := \mathbb{E}[\ell_\lambda(\theta, Z)]$ with $Z \sim \mu$, and the associated score generalization gap:

$$\mathscr{G}_\lambda(\mathbf{Z}^{(n)}, \theta) := \mathcal{R}^{(\lambda)}(\theta) - \widehat{\mathcal{R}}_{\mathbf{Z}^{(n)}}^{(\lambda)}(\theta) = \int\ell_\lambda(\theta, z)\mathrm{d}\mu(z) - \frac{1}{n}\sum_{i=1}^{n}\ell_\lambda(\theta, Z_i). \tag{9}$$

This definition of the generalization gap is consistent with practice as $\ell_\varpi$ is involved during training. Hence, upper bounding $\mathscr{G}_\lambda(\mathbf{Z}^{(n)}, \hat{\theta}^{(n)})$ is meaningful. Before exploring this route in Section 4, we first show that $\mathscr{G}_\lambda(\mathbf{Z}^{(n)}, \hat{\theta}^{(n)})$ naturally stems from $\varepsilon_{\mathrm{s}}^{(n)}$.

To state our next result, we define for any $n \in \mathbb{N}$, $(\overrightarrow{X}_t^{(n)})_{t\in[0,T]}$ as the solution of Equation (2) initialized randomly from the empirical distribution $\overrightarrow{X}_0^{(n)} \sim \hat{\mu}_n := n^{-1}\sum_{i=1}^{n}\delta_{Z_i}$ instead of $\mu$.

**Theorem 3.1.** *For all $\theta \in \Theta$, we have:*

$$\varepsilon_{\mathrm{s}}^{(n)}(\theta) = \mathscr{L}_{\mathrm{ESM}}^{(n)}(\theta) + \mathscr{G}_\lambda(\mathbf{Z}^{(n)}, \theta) + \widehat{\Delta}_T^{(n)},$$

*where $\widehat{\Delta}_T^{(n)} := \widehat{\mathrm{C}}_T^{(n)} - \mathrm{C}_T$,*

$$\widehat{\mathrm{C}}_T^{(n)} := \frac{4}{n}\sum_{i=1}^{n}\int\mathbb{E}[\|\nabla\log\tilde{p}_{t|0}(\overrightarrow{X}_t^{Z_i}|Z_i) - \nabla\log\tilde{p}_t^{(n)}(\overrightarrow{X}_t^{Z_i})\|^2]\mathrm{d}\lambda(t),$$

$$\mathrm{C}_T := 4 \int \mathbb{E}[\|\nabla \log \tilde{p}_t(\overrightarrow{X}_t^z) - \nabla \log \tilde{p}_{t|0}(\overrightarrow{X}_t^z|z)\|^2] \mathrm{d}(\mu \otimes \lambda)(z,t) \ ,$$

$$\mathscr{L}_{\mathrm{ESM}}^{(n)}(\theta) := \frac{h}{T} \sum_{k=0}^{N-1} \mathbb{E}\left[\|s_\theta(T-t_k, \overrightarrow{X}_{T-t_k}^{(n)}) - 2\nabla \log \tilde{p}_{T-t_k}^{(n)}(\overrightarrow{X}_{T-t_k}^{(n)})\|^2 | \mathbf{Z}^{(n)}\right] \ , \quad (10)$$

where we denote by $\tilde{p}_t^{(n)}$ the density of $\overrightarrow{X}_t^{(n)}$ with respect to $\gamma^d$.

The proof is postponed to Appendix A. We refer to the quantity $\widehat{\Delta}_T^{(n)}$ as the 'data-dependent diffusion gap' as it measures the discrepancy between the forward diffusions that are initialized at either the empirical data and the true data distribution. In addition, the term $\mathscr{L}_{\mathrm{ESM}}^{(n)}(\theta)$ is called the explicit score matching loss [Vin11]. It corresponds to the quality of the approximation of the empirical score $\nabla \log \tilde{p}_t^{(n)}$ by the score network, and is optimized by the learning algorithm.

Combining Theorem 2.1 with Theorem 3.1 implies that the control of the generative performance of $\nu_T^{(n)}$ boils down bounding $\widehat{\Delta}_T^{(n)}$ and $\mathscr{G}_\lambda(\mathbf{Z}^{(n)}, \theta)$. The former is handled in Section 3.2, quantifying the impact of $n$ via concentration arguments, and the latter in Section 4.

## 3.2 Quantifying the influence of the dataset size in generalization

We aim to upper-bound the term $\widehat{\Delta}_T^{(n)}$ of Theorem 3.1. To do so, we make the following assumption.

**Assumption 3.1.** *The data distribution $\mu$ has bounded support included in $\mathrm{B}(0, D)$, for some $D > 0$.*

This assumption is relatively common in the literature, and we typically expect it to be satisfied for image datasets. We expect that most of our proofs would still hold with assumptions on the moments of $\mu$. We leave these considerations for future works.

An important remark is that $\widehat{\Delta}_T^{(n)}$ is the difference between an empirical average and its theoretical counterpart. Based on this observation, we aim at quantifying the influence of $n$ in the generalization phenomenon. A first step is provided the following result.

**Lemma 3.1.** *Under Assumptions 2.1 and 3.1, with probability at least $1 - \delta$ over $\mathbf{Z}^{(n)} \sim \mu^{\otimes n}$,*

$$\widehat{\Delta}_T^{(n)} \leqslant 4D^2 \sqrt{\frac{\log(1/\delta)}{2n}} \int \mathrm{e}^{-2t} \mathrm{d}\lambda(t) + 4 \int \Delta \mathscr{I}_t^{(n)} \mathrm{d}\lambda(t) \ ,$$

*where $\Delta \mathscr{I}_t^{(n)} := \mathscr{I}(\overrightarrow{p}_t | \gamma^d) - \mathscr{I}(\overrightarrow{p}_t^{(n)} | \gamma^d)$ and $\lambda$ is defined in (8).*

Note first that $\widehat{\Delta}_T^{(n)}$ has a small contribution to the error if $t \mapsto \Delta \mathscr{I}_t^{(n)}$ stays negative over a subset of $[0, T]$ with large Lebesgue measure. A precise understanding of this phenomenon is a promising research direction, which we leave for future works.

To contextualize the previous remark, we rely on an observation that was also made by [FRDD25], in a submission that is parallel to our work. The next lemma shows that the data-dependent gap is negative in expectation over the dataset.

**Lemma 3.2.** *We have $\mathbb{E}\big[\widehat{\Delta}_T^{(n)}\big] \leqslant 0$, where the expectation is over $\mathbf{Z}^{(n)} \sim \mu^{\otimes n}$.*

This shows that, *in expectation*, the data-dependent gap $\widehat{\Delta}_T^{(n)}$ can be discarded from our decomposition. However, in our work, we are mainly interested in proving *high-probability bounds*, which we believe is essential as we expect to have an important variance in certain settings, as we make more precise in the following proposition.

We provide in the next proposition a quantitative bound on $\widehat{\Delta}_T^{(n)}$.

**Proposition 3.1.** *Under Assumption 3.1, with probability at least $1 - 2\delta$ over, we have: $\mathbf{Z}^{(n)} \sim \mu^{\otimes n}$:*

$$\widehat{\Delta}_T^{(n)} \lesssim \left(D^2 + K_1^2\right) \sqrt{\frac{\log(1/\delta)}{2n}} + \frac{h}{T} \mathscr{I}(\mu | \gamma^d) + K_1^2 \frac{\log(1/\delta)}{n} + \frac{\mathrm{W}^2 + K_2 \sqrt{h}\mathrm{W}}{Th} \ ,$$

*with $K_2^2 := D^2 + d\log(T/h) + hd$, $\quad K_1^2 := d(1 - \mathrm{e}^{-2h})^{-1} + D^2$, $\quad \mathrm{W} := \mathrm{W}_2\left(\overrightarrow{p}_{h/2}, \overrightarrow{p}_{h/2}^{(n)}\right)$.*

Proposition 3.1 does not provide an explicit convergence rate in $n$. The Fisher information term is the same as in Theorem 2.1 and is controlled when $N = T/h$ is large. On the other hand, when $h$ is fixed, the quantity W satisfies $\mathrm{W}^2 \leqslant \mathrm{e}^{-h}\mathrm{W}_2^2\left(\mu * \mathrm{N}(0, \bar{\sigma}^2\mathrm{I}_d), \widehat{\mu}_n * \mathrm{N}(0, \bar{\sigma}^2\mathrm{I}_d)\right)$, where $\bar{\sigma} := \sqrt{\mathrm{e}^h - 1}$, and the right-hand side corresponds to the smoothed Wasserstein distance between $\mu$ and $\hat{\mu}_n$. Bounding such quantities has received increasing attention in the literature [NGK21, BJPR25, GGNWP20]. In particular, [GGNWP20] proved that $\mathbb{E}\left[\mathrm{W}^2\right] = \mathcal{O}\left(n^{-1}\exp(2D^2/(\mathrm{e}^h - 1))\right)$. High-probability bounds with similar constants and $n$-dependence have also been established [NGK21]. Therefore, Proposition 3.1 implies that for a fixed $h$, a convergence rate of $\mathcal{O}\left(n^{-1/2}\right)$ can be achieved, albeit with constants that can grow rapidly as $h \to 0$. This quantifies the influence of dataset size on the generalization ability of SGMs. It can be seen that in worst case scenarios, plugging this bound in Theorem 2.1 and optimizing over $h$ leads to a rate of $n^{-\mathcal{O}(1/d)}$, similar to existing works [ADR24, Øks03].

In summary, combining Theorem 2.1, Theorem 3.1, and Proposition 3.1 yields a full characterization of the generalization error $\mathrm{KL}(\mu \mid \nu_T^{(n)})$, up to the score generalization gap $\mathscr{G}_\lambda(\mathbf{Z}^{(n)}, \theta)$, which we analyze next.

# 4 Unveiling the influence of the learning algorithm on generalization

We analyze next the score generalization gap $\mathscr{G}_\lambda(\mathbf{Z}^{(n)}, \theta^{(n)})$ when $\theta^{(n)}$ is the output of two algorithms: *(i)* the stochastic gradient Langevin dynamics [WT11] and *(ii)* the ADAM algorithm [KB17]. In both cases, the proposed results build upon existing generalization bounds. To our knowledge, this is the first time such a transfer of theoretical tools from supervised learning to generative modeling is possible, highlighting the benefits of the optimization algorithm, as a measure of the generalization error of diffusion models.

## 4.1 Experimental Setup

We set the Ornstein–Uhlenbeck process as our forward diffusion and use the cosine noise schedule [DN21]. We opt for the denoising parameterization of the model and its associated '$\epsilon$-loss' (see Appendix C, (22)), as introduced in [HJA20b] and widely used afterwards [DN21, HS21, SH22, KAAL22b, EKB+24]. This improves numerical stability and yields faster convergence as the high variance of the DSM loss (5) incurs noisier gradients and losses. As a result, we plot the generalization error as the difference between the train and test $\epsilon$-loss, which serves as a proxy for the score generalization error, as they only differ by a time-dependent multiplicative factor [SSDK+21]. We employ the Euler EI (6) to sample from our trained models. Full experimental details are available in Appendix C.

The first set of experiments is based on a $4$-dimensional dataset consisting of a mixture of 9 Gaussian distributions, with random means, and with some class imbalance to make the learning task harder[2]. Supplementary details are given in Appendix C.1. We then shift focus to higher-dimensional datasets, the `flowers` dataset [NZ06], and the `butterflies` dataset [WME09]. We also include experiments on the `MNIST` digits in Appendix C.3. For images, we use the DDPM++ U-Net architecture as implemented in [DN21]. We also study topological generalization bounds [ADS+24] associated to the training trajectories of the ADAM optimizer [KB17], which is the optimizer used in practice for state of the art models [RBL+22, EKB+24]. We vary learning rates and batch sizes to obtain multiple measure points and validate our bounds. Supplementary details are given in Appendix C.2.

## 4.2 Stochastic gradient Langevin dynamics and the influence of gradient norms

We first study the stochastic gradient Langevin dynamics (SGLD) [WT11], which we see as a noisy variant of SGD. In learning theory, the generalization ability of SGLD has been widely studied [RRT17, PJL18, FR21, NHD+19, DS24]. We define SGLD by the following recursion:

$$\theta_{k+1} = (1 - a\eta_k)\theta_k - \eta_k\widehat{g}_k(\theta) + \sqrt{2\eta_k\beta^{-1}}\mathrm{G}_k,$$

where $\mathrm{G}_k \sim \mathrm{N}(0, I_d)$, $\widehat{g}_k$ is an unbiased estimate of the gradient of the empirical risk, and $a \geqslant 0$ is a regularization coefficient. The term $\beta$ is called the inverse temperature parameter.

---

[2]The exact mixture weights are $(0.01, \ 0.1, \ 0.3, \ 0.2, \ 0.02, \ 0.15, \ 0.02, \ 0.15, \ 0.05)$

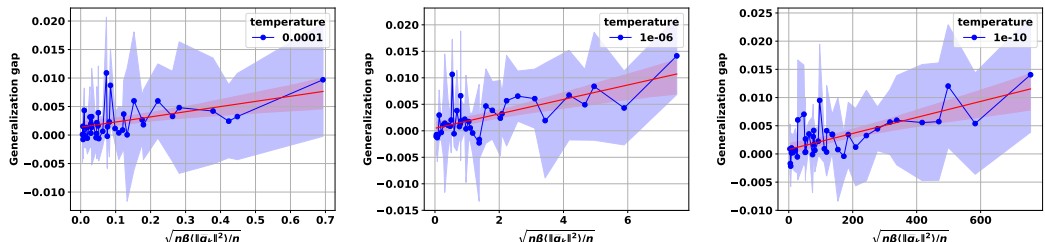

Figure 2: SGLD optimizer (17) on a low dimensional Gaussian mixture dataset, for different value of the temprature $(1/\beta)$. We use full batch size, constant learning rate $\eta$, a grid of values of $(n, \eta)$ and 10 random seeds. x-axis: Value of $\sqrt{\eta\beta\langle\|\widehat{g}_k^2\|\rangle/n}$. y-axis: Score generalization gap.

In the following, we fix a number of iterations $K \in \mathbb{N}^\star$ and denote by $\theta_K$ the output of SGLD after $K$ steps. To analyze the term $\mathcal{G}(\mathbf{Z}^{(n)}, \theta_K)$ in the case of SGLD, a wide variety of generalization bounds are available [DS24, Table 4], often involving expected gradient norms of the training process [MWZZ18, NHD+19, NDHR21, HNK+20]. Here, we exploit the seminal result of [MWZZ18] in the context of SGMs. First, recall that a random variable $X$ is $\tau^2$-subgaussian if for any $\alpha \in \mathbb{R}$, $\mathbb{E}\left[\exp(\alpha(X - \mathbb{E}[X]))\right] \leqslant \exp(\alpha^2\tau^2/2)$ [Ver18].

**Theorem 4.1** ([MWZZ18])**.** *We assume that for any $w \in \mathbb{R}^d$, the loss $\ell_\lambda(w, Z)$ is $\tau^2$-subgaussian with respect to $Z \sim \mu$ and that $\sup_k(\eta_k a) < 1$. We also assume the algorithm is initialized with $\theta_0 \sim \pi_0 = \mathrm{N}\left(0, \sigma_0^2 I_d\right)$ with $\sigma_0\sqrt{\beta a} \leqslant \sqrt{2}$. Then, with probability at least $1 - \delta$ over $\mathbf{Z}^{(n)} \sim \mu^{\otimes n}$, we have:*

$$\mathbb{E}\left[\mathcal{G}_\lambda(\mathbf{Z}^{(n)}, \theta_N)|\mathbf{Z}^{(n)}\right] \lesssim \frac{2\tau}{\sqrt{n}}\left\{\frac{\beta}{2}\sum_{k=0}^{K-1}\eta_k e^{-\frac{a}{2}(S_K - S_k)}\mathbb{E}[\|\widehat{g}_k\|^2] + \log\frac{3}{\delta}\right\}^{1/2}, \quad S_k := \sum_{j=0}^{k-1}\eta_j.$$

Theorem 4.1 shows that, up to a multiplicative constant involving $n$, the averaged gradient norms form an upper bound on $\mathcal{G}_\lambda(\mathbf{Z}^{(n)}, \theta_K)$ and thus impacts the generalization error of the model. To verify this claim, we consider the case of constant learning rates, *i.e.*, $\eta_k = \eta$, and take $n = 8192$, $a = 0$ and a batch size equal to $n$. Let $\langle\|\widehat{g}_k\|^2\rangle$ be the average gradient norm all the iterations. In Figure 2, obtained with SGLD on a low-dimensional gaussian mixture model, we compare the score generalization gap to the value of $B(n, \eta) := \sqrt{\eta\beta\langle\|\widehat{g}_k^2\|\rangle/n}$ for different inverse temperatures $\beta$, a grid of values of $n$ and $\eta$, and 10 random seeds. The order of magnitude of $B(n, \eta)$ in Figure 2 is bigger than the observed score generalization gap. This behavior is commonly seen for gradient-based bounds [DDS23] and may come from the unknown subgaussian constant $\tau$ in Theorem 4.1 or additional implicit regularization. Yet, the results support Theorem 4.1, reporting a good correlation between $B(n, \eta)$ and the generalization error, especially for high values of the inverse temperature $\beta$. While our theory does not take explicitly into account the model architecture, it is implicitly impacting the bound through the gradient norms in Theorem 4.1.

While our theory does not rigorously apply to more practical optimizers like SGD or ADAM because of the absence of Gaussian noise, we use the following heuristics based on [MHB16] to extend our experiments beyond the class of noisy algorithms. By considering that the variance of the stochastic gradient is of order $1/b$, where $b$ the batch size, we replace $\beta$ by $b/\eta$ and propose to compare the generalization error to $b\langle\|\widehat{g}_k\|^2\rangle$ and apply it to the ADAM optimizer. In this last case, we only average the last 200 gradients of training, to avoid noisy gradients in the first observation and characterize the geometry of the local minimum the model converged to. We observe in Figure 3 that this quantity correlates very well with the generalization error on the `butterflies` and `flowers` datasets. We provide in the appendix an additional experiment on a dataset (`MNIST`) with more data points, where the correlation is strong for most hyperparameters. A refined analysis shows that the observed correlation is related to the train loss of the model, suggesting that the relevance of the experiment increases near convergence, see Figures 5 and 6. Thus, our results suggest that gradient norms are a pertinent indicator of generalization for SGMs.

The question arises, as to whether the subgaussian assumption made in Theorem 4.1 can be met in practice. Verifying this assumption depends simultaneously on the *dataset* and the *architecture*, as it refers to the denoising score matching loss. While it cannot be seen as a direct consequence of our other assumptions, we expect that it can be satisfied in simple settings, such as image datasets or finitely supported distributions. We leave these considerations for future works.

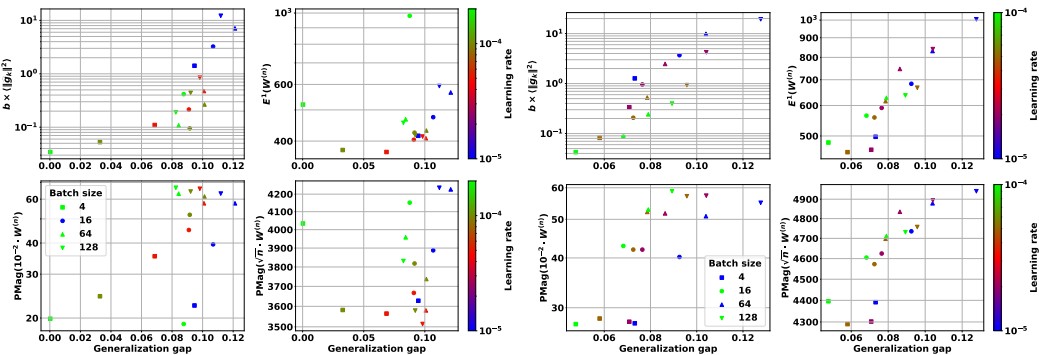

Figure 3: ADAM optimizer on the `butterflies` dataset *(left)* and the `flowers` dataset *(right)*. Generalization gap vs. several complexity metrics: $b\langle\|\widehat{g}_k\|^2\rangle$ *(top left)*, $E^1(\mathcal{W}^{(n)})$ *(top right)*, $\text{PMag}(10^{-2} \cdot \mathcal{W}^{(n)})$ *(bottom left)* and $\text{PMag}(\sqrt{n} \cdot \mathcal{W}^{(n)})$ *(bottom right)*.

## 4.3 The influence of training trajectories and application to ADAM

While providing fruitful insights and generalization measures, SGLD might be far from the learning algorithms used in practice (e.g., ADAM). Here, we exploit the recent *topological* generalization bounds of [DVDS24, ADS$^+$24], which can be applied for a large class of algorithms including ADAM. Topological bounds are based on the intuition that the *training trajectory* (*i.e.*, the parameter sequence generated by the optimization procedure) might encode topological properties of local minima, related to their generalization ability.

We build our analysis on the results of [ADS$^+$24]. Let us fix $k_0, k_1 \in \mathbb{N}^\star$ and introduce the training trajectory $\mathcal{W}^{(n)} := \{\hat{\theta}_k^{(n)}, \ k_0 \leqslant k \leqslant k_1\}$, where $\hat{\theta}_k^{(n)}$ denotes the learned parameter of the score network at the $k$-th iteration, and $k_0$ is chosen such that $\hat{\theta}_k^{(n)}$ is close to a local minimum (near convergence). Topological bounds relate the generalization error to quantities quantifying the topological complexity of $\mathcal{W}^{(n)}$, stemming from topological data analysis (TDA) [BCY18]. We focus here on the particular case where this complexity is the *weighted lifetime sum* [Sch20], which informally tracks down the number of clusters of $\mathcal{W}^{(n)}$ at different scales. We denote it $E^1(\mathcal{W}^{(n)})$ and formally introduce it in Appendix B.2. More precisely, $E^1(\mathcal{W}^{(n)})$ denotes the cost of a minimum spanning tree over the finite set $\mathcal{W}^{(n)}$, for a given distance (e.g., the Euclidean distance). We refer to [ADS$^+$24] for an introduction of this notion in the context of generalization bounds.

The next theorem shows how the weighted lifetime sum upper-bounds the generation performance.

**Theorem 4.2** ([ADS$^+$24])**.** *Assume that the loss $(\theta, z) \mapsto \ell_\lambda(\theta, z)$ is uniformly bounded by $B > 0$. Suppose Assumption 2.1. Then, with probability at least $1 - \delta$, we have for all $\theta \in \mathcal{W}^{(n)}$ that:*

$$\mathscr{G}_\lambda(\mathbf{Z}^{(n)}, \theta) \lesssim B\sqrt{\frac{\log(1 + (4\sqrt{n}/B)E^1(\mathcal{W}^{(n)})) + 1 + I_\infty(\mathcal{W}^{(n)}, \mathbf{Z}^{(n)}) + \log(1/\delta)}{n}},$$

*where $I_\infty(\mathcal{W}^{(n)}, \mathbf{Z}^{(n)})$ is a mutual information term defined in Appendix B.2.1.*

Note that a similar bound involving another notion of complexity (the *positive magnitude*) is presented in Theorem B.1, due to space limitations. The positive magnitude, introduced by [ADS$^+$24], is a quantity of similar flavor to $E^1(\mathcal{W}^{(n)})$ and additionally depends on a scale parameter $r > 0$, it is denoted $\text{PMag}(r \cdot \mathcal{W}^{(n)})$. In our experiments, we consider the choice $r = \sqrt{n}$ (which is theoretically justified by [ADS$^+$24]) and $r = 10^{-2}$, as these authors argued that positive magnitude for smaller values of $r$ empirically correlates with the generalization error (the scale $10^{-2}$ is used by these authors).

The information-theoretic term in Theorems 4.2 and B.1 is hard to estimate in practice, even though it was successfully bounded in particular cases [DVDS24]. For this reason, we proceed as in [ADS$^+$24] and empirically illustrate the correlation between the three topological complexities ($E^1(\mathcal{W}^{(n)})$, $\text{PMag}(10^{-2} \cdot \mathcal{W}^{(n)})$ and $\text{PMag}(\sqrt{n} \cdot \mathcal{W}^{(n)})$) and the score generalization gap in Figure 3. Up to our knowledge, it is the first time that these topological generalization bounds are evaluated for diffusion models. For the `butterflies` and `flowers` datasets, we observe in Figure 3 that the three

proposed topological complexities correlate very well with the score generalization gap. Slightly worse correlations are observed for $\text{PMag}(10^{-2}\mathcal{W}^{(n)})$, which is in line with the theory of [ADS+24]. We provide some generated image grids for the `butterfly` dataset in Figure 4 for visual comparison. Table 2 reports Train/Test DSM together with $b \times \langle\|\widehat{g}\|^2\rangle$, $E^1(\mathcal{W}^{(n)})$, and $\text{PMag}(\cdot)$ across learning rates and batch sizes for ADAM on `flowers`, complementing Fig. 3.

We also include additional experiments for the `MNIST` dataset in the appendix, see Figures 5 and 6, In that case, the positive magnitude also has satisfying correlation with the generalization error while for $E^1$ the situation is slightly more contrasted. Similar to the above, it seems the lack of convergence of the model can negatively impact the relevance of $E^1$, which is coherent with the observations of [DDS23, ADS+24]. We also make the new observation that $E^1$ and the gradient norms-based bounds have very similar behavior. Thus, our experiments show that the topology of the training trajectories has an impact on the generalization error of SGMs.

Finally, let us note that our generalization bounds do not explicitly take into account the noise schedule, which is known to empirically affect the performance of the score network, but, as the noise schedule affects the properties of the optimization dynamics, we expect its effect to be captured by the generalization error term in our decomposition.

# 5 Conclusion

In this paper, we proposed an algorithm- and data-dependent analysis of the generalization abilities of practically used diffusion models. Our theoretical analysis is based on a decomposition of the score approximation error. After providing two upper bounds of a statistical ersatz arising from this approach, we focus our discussion on what we call the score generalization gap, which represents the generalization error associated to the denoising score matching loss used during training. We apply our framework to several classical stochastic optimization algorithms and obtain generalization bounds with explicit dependence on the training dynamics. These results altogether yielded a KL bound with $\mathcal{O}(n^{-1/2})$ rate. Based on these observation, we numerically evaluated the correlation between the score generalization gap and the two topological complexity measures and gradient norms, hence, providing new empirical insights for diffusion models.

**Limitations and future works.** Our theoretical bound in Section 3.2 may be coarse in certain scenarios, and overall suggests potential refinements incorporating information-theoretic quantities such as the conditional entropy of the data given its noisy observation, as inspired by recent work on entropy-based noise schedules. This could lead to a deeper understanding of generalization, particularly regarding more involved data-dependent quantities. On the experimental side, thorough evaluation of our theoretical predictions requires well-trained diffusion models across a wide range of settings. However, the high computational cost of full training runs, while varying many hyperparameters, currently limits the scale of our empirical analysis. In particular, we leave for future work a broader exploration involving higher-dimensional datasets and larger training sets.

# Acknowledgments

U.S. is partially supported by the French government partly funded this work under the management of Agence Nationale de la Recherche as part of the "France 2030" program, reference ANR-23-IACL-0008 (PR[AI]RIE-PSAI). B.D., M.H., D.S., and U.S. are partially supported by the European Research Council Starting Grant DYNASTY – 101039676. A.D. acknowledges funding by the European Union (ERC-2022-SyG, 101071601). Views and opinions expressed are however those of the authors only and do not necessarily reflect those of the European Union or the European Research Council Executive Agency. Neither the European Union nor the granting authority can be held responsible for them. The authors thank Tyler Farghly and Eliot Beyler for helpful discussions. The authors are grateful to the CLEPS infrastructure from the Inria of Paris for providing resources and support. This work was granted access to the HPC resources of IDRIS under the allocation 2025-AD011015323R1 made by GENCI.

**Broader impact statement.** Our research is theoretical and raises no direct societal or ethical concerns.

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

- Finally, in Appendix C, we provide the full details of our experimental setup, discuss some additional empirical results, and finally offer some final remarks regarding extensions to other transport-based generative models.

# A Omitted proofs of Section 3

Given a fixed dataset $\mathbf{Z}^{(n)} := (Z_1, \ldots, Z_n) \sim \mu^{\otimes n}$, we will frequently use the notation:

$$\widehat{\mu}_n := \frac{1}{n} \sum_{i=1}^{n} \delta_{Z_i}. \tag{11}$$

We also recall that we denote $\overrightarrow{p}_t$ the density of $\overrightarrow{X}_t$ with respect to the Lebesgue measure (where $\overrightarrow{X}_t$ is initialized from $\mu$) and $\tilde{p}_t$ its density with respect to the Gaussian measure $\gamma^d$. Similarly, we denote $\overrightarrow{X}_t^{(n)}$ the process following Equation (2) initialized from the empirical distribution $\widehat{\mu}_n$. For $t > 0$, we denote by $\overrightarrow{p}_t^{(n)}$ its density with respect to the Lebesgue measure and by $\tilde{p}_t^{(n)}$ its density with respect to $\gamma^d$. Finally, $\overrightarrow{p}_{t|0}$ is the density of $\tilde{X}$ given $\overrightarrow{X}_0$ with respect to the Lebesgue measure and $\tilde{p}_{t|0}$ its density with respect to $\gamma^d$.

We start by a technical lemma which is taken from the proof of Equation (11) in [Vin11], which we reprove with our notations for the sake of completeness.

**Lemma A.1.** *Consider a probability measure $\nu$ on $\mathbb{R}^d$. Only for this lemma, we denote by $\tilde{p}_t$ the density with respect to $\gamma^d$ of the process $\overrightarrow{X}_t$ initialized from $\overrightarrow{X}_t \sim \nu$. For any Borel-measurable function $\psi : \mathbb{R}^d \to \mathbb{R}^d$ and fixed time $t > 0$, we have the following identity:*

$$\mathbb{E}\left[\langle \psi(\overrightarrow{X}_t), \nabla \log \tilde{p}_t(\overrightarrow{X}_t)\rangle\right] = \mathbb{E}\left[\langle \psi(\overrightarrow{X}_t), \nabla \log \tilde{p}_{t|0}(\overrightarrow{X}_t|\overrightarrow{X}_0)\rangle\right].$$

*In particular, this lemma can be written as:*

$$\int \mathbb{E}\left[\langle \psi(\overrightarrow{X}_t^z), \nabla \log \tilde{p}_t(\overrightarrow{X}_t^z)\rangle\right] \mathrm{d}\nu(z) = \int \mathbb{E}\left[\langle \psi(\overrightarrow{X}_t^z), \nabla \log \tilde{p}_{t|0}(\overrightarrow{X}_t^z|z)\rangle\right] \mathrm{d}\nu(z).$$

*Proof.* Let $Z \sim \mu$, by Fisher's identity and the tower property for conditional expectation, we have:

$$
\begin{aligned}
\mathbb{E}\left[\langle \psi(\overrightarrow{X}_t), \nabla \log \tilde{p}_t(\overrightarrow{X}_t)\rangle\right] &= \mathbb{E}\left[\langle \psi(\overrightarrow{X}_t), \mathbb{E}\left[\nabla \log \tilde{p}_{t|0}(\overrightarrow{X}_t|\overrightarrow{X}_0)|\overrightarrow{X}_t\right]\rangle\right] \\
&= \mathbb{E}\left[\mathbb{E}\left[\langle \psi(\overrightarrow{X}_t), \nabla \log \tilde{p}_{t|0}(\overrightarrow{X}_t|\overrightarrow{X}_0)\rangle|\overrightarrow{X}_t\right]\right] \\
&= \mathbb{E}\left[\langle \psi(\overrightarrow{X}_t), \nabla \log \tilde{p}_{t|0}(\overrightarrow{X}_t|\overrightarrow{X}_0)\rangle\right].
\end{aligned}
$$

□

## A.1 Proof of Theorem 3.1.

In this subsection, we present the proof of Theorem 3.1 The proof relies on classical computations on score functions [Vin11, OAS23]. We start with the following lemma, which provides a decomposition of the score approximation in terms of the denoising score matching loss.

**Lemma A.2.** *For all $\theta \in \Theta$, we have $\varepsilon_s^{(n)}(\theta) = \mathscr{L}_{\mathrm{DSM}}^{(n)}(\theta) + \mathscr{G}_\lambda(\mathbf{Z}^{(n)}, \theta) - \mathrm{C}_T$, where $\mathrm{C}_T \geqslant 0$ is a non-negative constant (independent of $\theta$) defined by:*

$$\mathrm{C}_T := 4 \int \mathbb{E}\left[\left\|\nabla \log \tilde{p}_t(\overrightarrow{X}_t^z) - \nabla \log \tilde{p}_{t|0}(\overrightarrow{X}_t^z|z)\right\|^2\right] \mathrm{d}(\mu \otimes \lambda)(x, t), \tag{12}$$

*with $\lambda := T^{-1} \sum_{k=0}^{N-1} h\delta_{T-t_k}$.*

*Proof.* Let's recall that $\overrightarrow{X}_t$ denotes a solution of Equation (2) initialized with $\overrightarrow{X}_0 \sim \mu$, where $\mu$ is the data distribution. Let us recall the definition of the probability measure $\lambda$:

$$\lambda := \frac{1}{T} \sum_{k=0}^{N-1} h_{k+1} \delta_{T-t_k}.$$

Note that the support of $\lambda$ is bounded away from 0, which justifies the derivations below.

We expand the square and use Lemma A.1 to obtain:

$$\begin{aligned}
\varepsilon_{\mathrm{s}}(\theta) &= \int \mathbb{E}\left[\left\|s_\theta(t, \overrightarrow{X}_t) - 2\nabla \log \tilde{p}_t(\overrightarrow{X}_t)\right\|^2\right] \mathrm{d}\lambda(t) \\
&= \int \left(\mathbb{E}\left[\left\|s_\theta(t, \overrightarrow{X}_t)\right\|^2\right] + 4\mathbb{E}\left[\left\|\nabla \log \tilde{p}_t(\overrightarrow{X}_t)\right\|^2\right]\right) \mathrm{d}\lambda(t) \\
&\quad - 4\int \mathbb{E}\left[\left\langle s_\theta(t, \overrightarrow{X}_t), \nabla \log \tilde{p}_t(\overrightarrow{X}_t)\right\rangle\right] \mathrm{d}\lambda(t) \\
&= \int \left(\mathbb{E}\left[\left\|s_\theta(t, \overrightarrow{X}_t)\right\|^2\right] + 4\mathbb{E}\left[\left\|\nabla \log \tilde{p}_t(\overrightarrow{X}_t)\right\|^2\right]\right) \mathrm{d}\lambda(t) \\
&\quad - 4\int \int \mathbb{E}\left[\left\langle s_\theta(t, \overrightarrow{X}_t^z), \nabla \log \tilde{p}_{t|0}(\overrightarrow{X}_t^z|z)\right\rangle\right] \mathrm{d}\mu(z)\mathrm{d}\lambda(t) \\
&= \int \int \mathbb{E}\left[\left\|s_\theta(t, \overrightarrow{X}_t^z) - 2\nabla \log \tilde{p}_{t|0}(\overrightarrow{X}_t^z|z)\right\|^2\right] \mathrm{d}\mu(z)\mathrm{d}\lambda(t) \\
&\quad - 4\int \left(\int \mathbb{E}\left[\left\|\nabla \log \tilde{p}_{t|0}(\overrightarrow{X}_t^z|z)\right\|^2\right] \mathrm{d}\mu(z) - \mathbb{E}\left[\left\|\nabla \log \tilde{p}_t(\overrightarrow{X}_t)\right\|^2\right]\right) \mathrm{d}\lambda(t) \\
&= \mathcal{R}^{(\lambda)}(\theta) - 4\int \left(\int \mathbb{E}\left[\left\|\nabla \log \tilde{p}_{t|0}(\overrightarrow{X}_t^z|z)\right\|^2\right] \mathrm{d}\mu(z) - \mathbb{E}\left[\left\|\nabla \log \tilde{p}_t(\overrightarrow{X}_t)\right\|^2\right]\right) \mathrm{d}\lambda(t),
\end{aligned}$$

The derivation above is identical to Lemma C.3 in [OAS23] and is a direct consequence of the celebrated result of [Vin11].

Now, we note that $\mathcal{R}^{(\lambda)}(\theta) = \widehat{\mathcal{R}}_{\mathbf{Z}^{(n)}}^{(\lambda)}(\theta) + \mathcal{G}(\mathbf{Z}^{(n)}, \theta) = \mathcal{L}_{\mathrm{DSM}}^{(n)}(\theta) + \mathcal{G}(\mathbf{Z}^{(n)}, \theta)$ by definition of the denoising score matching loss. Therefore, we conclude the proof of Lemma A.2 by using the following lemma. $\square$

**Lemma A.3.** *We have the following identity:*

$$\frac{C_T}{4} = \int \left(\int \mathbb{E}\left[\left\|\nabla \log \tilde{p}_{t|0}(\overrightarrow{X}_t^z|z)\right\|^2\right] \mathrm{d}\mu(z) - \mathbb{E}\left[\left\|\nabla \log \tilde{p}_t(\overrightarrow{X}_t)\right\|^2\right]\right) \mathrm{d}\lambda(t).$$

*Proof.* We just need to show that $C_T \geqslant 0$, we see it by the following calculations based on Lemma A.1. We have:

$$\begin{aligned}
\frac{C_T}{4} &= \int \left(\mathbb{E}\left[\left\|\nabla \log \tilde{p}_t(\overrightarrow{X}_t)\right\|^2\right] + \int \mathbb{E}\left[\left\|\nabla \log \tilde{p}_{t|0}(\overrightarrow{X}_t^z|z)\right\|^2\right] \mathrm{d}\mu(z)\right) \mathrm{d}\lambda(t) \\
&\quad - 2\int \int \mathbb{E}\left[\left\langle \nabla \log \tilde{p}_t(\overrightarrow{X}_t), \nabla \log \tilde{p}_{t|0}(\overrightarrow{X}_t^z|z)\right\rangle\right] \mathrm{d}\mu(z)\mathrm{d}\lambda(t) \\
&= \int \left(\mathbb{E}\left[\left\|\nabla \log \tilde{p}_t(\overrightarrow{X}_t^z)\right\|^2\right] + \int \mathbb{E}\left[\left\|\nabla \log \tilde{p}_{t|0}(\overrightarrow{X}_t^z|z)\right\|^2\right] \mathrm{d}\mu(z)\right) \mathrm{d}\lambda(t) \\
&\quad - 2\int \mathbb{E}\left[\left\langle \nabla \log \tilde{p}_t(\overrightarrow{X}_t), \nabla \log \tilde{p}_t(\overrightarrow{X}_t)\right\rangle\right] \mathrm{d}\lambda(t) \\
&= \int \left(\int \mathbb{E}\left[\left\|\nabla \log \tilde{p}_{t|0}(\overrightarrow{X}_t^z|z)\right\|^2\right] \mathrm{d}\mu(z) - \mathbb{E}\left[\left\|\nabla \log \tilde{p}_t(\overrightarrow{X}_t)\right\|^2\right]\right) \mathrm{d}\lambda(t).
\end{aligned}$$

This completes the proof of Lemma A.2. $\square$

An immediate consequence of Lemma A.2 is that $\varepsilon_{\mathrm{s}}^{(n)}(\theta) \leqslant \mathcal{L}_{\mathrm{DSM}}(\theta) + \mathcal{G}(\mathbf{Z}^{(n)}, \theta)$. Such a result is consistent with the minimisation of $\mathcal{L}_{\mathrm{DSM}}$ made in practice.

**Proof of Theorem 3.1.**

*Proof.* By definition, we have for a *fixed* $\mathbf{Z}^{(n)} = (Z_1, \ldots, Z_n) \in (\mathbb{R}^d)^n$ that:

$$\mathscr{L}_{\mathrm{DSM}}^{(n)}(\theta) = \frac{1}{n} \sum_{i=1}^{n} \int \mathbb{E} \left[ \left\| s_\theta(t, \overrightarrow{X}_t^{Z_i}) - 2\nabla \log \tilde{p}_{t|0}(\overrightarrow{X}_t^{Z_i}|Z_i) \right\|^2 \right] \mathrm{d}\lambda(t).$$

Therefore, we can apply Lemma A.1 to obtain:

$$\mathscr{L}_{\mathrm{DSM}}^{(n)}(\theta) = \frac{1}{n} \sum_{i=1}^{n} \int \left( \mathbb{E} \left[ \left\| s_\theta(t, \overrightarrow{X}_t^{Z_i}) \right\|^2 \right] + 4\mathbb{E} \left[ \left\| \nabla \log \tilde{p}_{t|0}(\overrightarrow{X}_t^{Z_i}|Z_i) \right\|^2 \right] \right.$$
$$\left. - 4\mathbb{E} \left[ \left\langle s_\theta(t, \overrightarrow{X}_t^{Z_i}), \nabla \log \tilde{p}_{t|0}(\overrightarrow{X}_t^{Z_i}|Z_i) \right\rangle \right] \right) \mathrm{d}\lambda(t)$$

$$= \frac{1}{n} \sum_{i=1}^{n} \int \left( \mathbb{E} \left[ \left\| s_\theta(t, \overrightarrow{X}_t^{Z_i}) \right\|^2 \right] + 4\mathbb{E} \left[ \left\| \nabla \log \tilde{p}_{t|0}(\overrightarrow{X}_t^{Z_i}|Z_i) \right\|^2 \right] \right.$$
$$\left. - 4\mathbb{E} \left[ \left\langle s_\theta(t, \overrightarrow{X}_t^{Z_i}), \nabla \log \tilde{p}_t^{(n)}(\overrightarrow{X}_t^{Z_i}) \right\rangle \right] \right) \mathrm{d}\lambda(t)$$

$$= \mathscr{L}_{\mathrm{ESM}}^{(n)}(\theta) + \frac{4}{n} \sum_{i=1}^{n} \int \left( \mathbb{E} \left[ \left\| \nabla \log \tilde{p}_{t|0}(\overrightarrow{X}_t^{Z_i}|Z_i) \right\|^2 \right] - \mathbb{E} \left[ \left\| \nabla \log \tilde{p}_t^{(n)}(\overrightarrow{X}_t^{Z_i}) \right\|^2 \right] \right) \mathrm{d}\lambda(t),$$

Thus, we have $\mathscr{L}_{\mathrm{DSM}}^{(n)}(\theta) = \mathscr{L}_{\mathrm{ESM}}^{(n)}(\theta) + \widehat{\mathrm{C}}_T^{(n)}$, by definition of $\widehat{\mu}_n$. We conclude by copying the proof of Lemma A.3 to obtain that:

$$\frac{\widehat{\mathrm{C}}_T^{(n)}}{4} = \frac{1}{n} \sum_{i=1}^{n} \left( \mathbb{E} \left[ \left\| \nabla \log \tilde{p}_{t|0}(\overrightarrow{X}_t^{Z_i}|Z_i) \right\|^2 \right] - \mathbb{E}_{\lambda, B, \widehat{\mu}_n} \left[ \left\| \nabla \log \tilde{p}_t^{(n)}(\overrightarrow{X}_t^{Z_i}) \right\|^2 \right] \right).$$

$\square$

## A.2 Omitted proofs of Section 3.2

In this subsection, we present the omitted proofs of Section 3.2.

**Proof of Lemma 3.1.**

*Proof.* Using the previous computations, we have $\widehat{\Delta}_T^{(n)} = \mathrm{I} + \mathrm{II}$, with:

$$\mathrm{I} := \int \left( \frac{1}{n} \sum_{i=1}^{n} 4\mathbb{E} \left[ \left\| \nabla \log \tilde{p}_{t|0}(\overrightarrow{X}_t^{Z_i}|Z_i) \right\|^2 \right] - \int 4\mathbb{E} \left[ \left\| \nabla \log \tilde{p}_{t|0}(\overrightarrow{X}_t^z|z) \right\|^2 \right] \mathrm{d}\mu(z) \right) \mathrm{d}\lambda(t),$$

$$\mathrm{II} := \int \left( 4\mathbb{E} \left[ \left\| \nabla \log \tilde{p}_t(\overrightarrow{X}_t) \right\|^2 \right] - \frac{4}{n} \sum_{i=1}^{n} \mathbb{E} \left[ \left\| \nabla \log \tilde{p}_t^{(n)}(\overrightarrow{X}_t^{Z_i}) \right\|^2 \right] \right) \mathrm{d}\lambda(t).$$

By Mehler's formula applied on the forward Ornstein-Uhlenbeck process, we know that $\overrightarrow{p}_{t|0}(\cdot|z)$ is the density of $\mathrm{N}(\mathrm{e}^{-t}z, (1 - \mathrm{e}^{-2t}) I_d)$. Therefore, we have that for any probability measure $\nu$:

$$4 \int \mathbb{E} \left[ \left\| \nabla \log \tilde{p}_{t|0}(\overrightarrow{X}_t^z|z) \right\|^2 \right] \mathrm{d}\nu(z)\mathrm{d}\lambda(t) = 4 \int \mathbb{E} \left[ \left\| -\frac{\overrightarrow{X}_t^z - \mathrm{e}^{-t}z}{1 - \mathrm{e}^{-2t}} + \overrightarrow{X}_t^z \right\|^2 \right] \mathrm{d}\nu(z)\mathrm{d}\lambda(t)$$

$$= \int \mathbb{E} \left[ \frac{4}{(\mathrm{e}^{2t} - 1)^2} \left\| \overrightarrow{X}_t^z - \mathrm{e}^t z \right\|^2 \right] \mathrm{d}\nu(z)\mathrm{d}\lambda(t)$$

$$= \int \frac{4}{(\mathrm{e}^{2t} - 1)^2} \left( 4\sinh^2(t) \|z\|^2 + d \left( 1 - \mathrm{e}^{-2t} \right) \right) \mathrm{d}\nu(z)\mathrm{d}\lambda(t)$$

$$= 4 \int \mathrm{e}^{-2t} \left( \|\nu\|^2 + \frac{d}{\mathrm{e}^{2t} - 1} \right) \mathrm{d}\lambda(t).$$

with $\|\nu\|^2 := \mathbb{E}_{x \sim \nu}\left[\|x\|^2\right]$. Therefore, we have:

$$\mathrm{I} = 4 \int \mathrm{e}^{-2t}\left(\|\widehat{\mu}_n\|^2 - \|\mu\|^2\right)\mathrm{d}\lambda(t).$$

We have that $\|\widehat{\mu}_n\|^2 - \|\mu\|^2 = \int \|Z\|^2\mathrm{d}\mu(Z) - \frac{1}{n}\sum_{i=1}^n \|Z_i\|^2$, with $(Z_1, \ldots, Z_n) \sim \mu^{\otimes n}$. By Assumption 3.1 we have that $\|Z_i\|^2 \leqslant D^2$ almost surely. Therefore, by Hoeffding's inequality, we have:

$$\mu^{\otimes n}\left(\int \|Z\|^2\mathrm{d}\mu(Z) - \frac{1}{n}\sum_{i=1}^n \|Z_i\|^2 \geqslant \epsilon\right) \leqslant \exp\left(-\frac{2n\epsilon^2}{D^4}\right),$$

from which we deduce that with probability at least $1 - \delta$ we have:

$$\mathrm{I} \leqslant 4D^2\sqrt{\frac{\log(1/\delta)}{2n}}\int \mathrm{e}^{-2t}\mathrm{d}\lambda(t).$$

Finally, we remark that by definition of the relative densities and Fisher information, we have:

$$\mathrm{II} = 4 \int \left(\int \left\|\nabla\log\frac{\overrightarrow{p}_t(y)}{\gamma^d(y)}\right\|^2 \overrightarrow{p}_t(y)\mathrm{d}y - \int \left\|\nabla\log\frac{\overrightarrow{p}_t^{(n)}(y)}{\gamma^d(y)}\right\|^2 \overrightarrow{p}_t(y)\mathrm{d}y\right)\mathrm{d}\lambda(t)$$

$$= 4 \int \left(\mathscr{I}(\overrightarrow{p}_t|\gamma^d) - \mathscr{I}(\overrightarrow{p}_t^{(n)}|\gamma^d)\right)\mathrm{d}\lambda(t).$$

This concludes the proof. $\qquad\square$

**Proof of Lemma 3.2.**

*Proof.* By definition, we have,

$$\widehat{\mathrm{C}}_T^{(n)} := 4 \int \mathbb{E}[\|\nabla\log\tilde{p}_{t|0}(\overrightarrow{X}_t^z|z) - \nabla\log\tilde{p}_t^{(n)}(\overrightarrow{X}_t^z)\|^2]\mathrm{d}(\widehat{\mu}_n \otimes \lambda)(z,t) ,$$

where we recall that $\widehat{\mu}_n := n^{-1}\sum_{i=1}^n \delta_{Z_i}$. By Lemma A.1, we have

$$\widehat{\mathrm{C}}_T^{(n)} = 4 \int \mathbb{E}[\|\nabla\log\tilde{p}_{t|0}(\overrightarrow{X}_t^z|z) - \nabla\log\tilde{p}_t(\overrightarrow{X}_t^z)\|^2]\mathrm{d}(\widehat{\mu}_n \otimes \lambda)(z,t)$$

$$- 4 \int \mathbb{E}[\|\nabla\log\tilde{p}_t^{(n)}(\overrightarrow{X}_t^z) - \nabla\log\tilde{p}_t(\overrightarrow{X}_t^z)\|^2]\mathrm{d}(\widehat{\mu}_n \otimes \lambda)(z,t).$$

By discarding the negative term and taking the expectation, we obtain

$$\mathbb{E}_{\mathbf{Z}^{(n)}}\left[\widehat{\mathrm{C}}_T^{(n)}\right] \leqslant 4\mathbb{E}_{\mathbf{Z}^{(n)}}\left[\int \mathbb{E}[\|\nabla\log\tilde{p}_{t|0}(\overrightarrow{X}_t^z|z) - \nabla\log\tilde{p}_t(\overrightarrow{X}_t^z)\|^2]\mathrm{d}(\widehat{\mu}_n \otimes \lambda)(z,t)\right]$$

$$= \mathrm{C}_T,$$

where the last equality follows from the definition of $\widehat{\mu}_n$. This concludes the proof. $\qquad\square$

Before proceeding to the proof of Proposition 3.1, we prove three intermediary lemmas.

First, we need a discretization lemma to control the measure $\lambda$ and the uniform measure on $[h, T]$.

**Lemma A.4.** *Let us introduce* $\Delta\mathscr{I}_t^{(n)} := \mathscr{I}(\overrightarrow{p}_t|\gamma^d) - \mathscr{I}(\overrightarrow{p}_t^{(n)}|\gamma^d)$. *Then, we have:*

$$\sum_{k=0}^{N-1} h\Delta\mathscr{I}_{T-t_k}^{(n)} - \int_h^T \Delta\mathscr{I}_t^{(n)}\mathrm{d}t \leqslant h\mathscr{I}(\overrightarrow{p}_{T-t_{N-1}}|\gamma^d).$$

*Proof.* Let $(P_t)_{t \geqslant 0}$ denote the Ornstein-Uhlenbeck semigroup associated with Equation (2). It is known that the Fisher information $t \mapsto \mathscr{I}(\mu P_t|\gamma^d)$ is a continuous function of time for $t > 0$ and is

decreasing along the Ornstein-Uhlenbeck semigroup. It can be seen for instance by noting that for $t > 0$.

$$\frac{\mathrm{d}}{\mathrm{d}t} \mathscr{I}(\mu P_t | \gamma^d) = -2 \int \phi_t \Gamma_2(\log \phi_t) \mathrm{d}\gamma^d,$$

where $\phi_t$ is the density of $\mu P_t = \overrightarrow{p}_t$ wrt $\gamma^d$ and $\Gamma_2$ is the iterated "carré du champ" operator [BGL14, Section 5.7]. Therefore, we have the following comparison between the sum and the integral.

$$\begin{aligned}
\sum_{k=0}^{N-1} h \mathscr{I}(\mu P_{T-t_k} | \gamma^d) &= h \mathscr{I}(\mu P_{T-t_{N-1}} | \gamma^d) + \sum_{k=0}^{N-2} \int_{T-t_{k+1}}^{T-t_k} \mathscr{I}(\mu P_{T-t_k} | \gamma^d) \mathrm{d}s \\
&\leqslant h \mathscr{I}(\mu P_{T-t_{N-1}} | \gamma^d) + \sum_{k=0}^{N-2} \int_{T-t_{k+1}}^{T-t_k} \mathscr{I}(\mu P_s | \gamma^d) \mathrm{d}s \\
&= h \mathscr{I}(\mu P_{T-t_{N-1}} | \gamma^d) + \int_{T-t_{N-1}}^{T} \mathscr{I}(\mu P_s | \gamma^d) \mathrm{d}s.
\end{aligned}$$

On the other hand, we have that:

$$\begin{aligned}
\sum_{k=0}^{N-1} h \mathscr{I}(\widehat{\mu}_n P_{T-t_k} | \gamma^d) &\geqslant \sum_{k=1}^{N-1} h \mathscr{I}(\widehat{\mu}_n P_{T-t_k} | \gamma^d) \\
&= \sum_{k=1}^{N-1} \int_{T-t_k}^{T-t_{k-1}} \mathscr{I}(\widehat{\mu}_n P_{T-t_k} | \gamma^d) \mathrm{d}s \\
&\geqslant \sum_{k=1}^{N-1} \int_{T-t_k}^{T-t_{k-1}} \mathscr{I}(\widehat{\mu}_n P_s | \gamma^d) \mathrm{d}s \\
&= \int_{T-t_{N-1}}^{T} \mathscr{I}(\widehat{\mu}_n P_s | \gamma^d) \mathrm{d}s.
\end{aligned}$$

Combining these inequalities, we immediately obtain the desired result by noting that for all $t > 0$ we have $\mu P_t = \overrightarrow{p}_t$, $\widehat{\mu}_n P_t = \overrightarrow{p}_t^{(n)}$ and by noting that $T - t_{N-1} = h$. $\qquad \square$

The next lemma provides a uniform bound on various expected score norms appearing in our proofs.

**Lemma A.5.** *Assume that $\mu$ has a support bounded by $D$. Consider positive numbers $0 < a < b$. Then, almost surely for $x \sim \mu$ (or $x \sim \widehat{\mu}_n$ and $\mathbf{Z}^{(n)} \sim \mu^{\otimes n}$), we have:*

$$\frac{1}{b-a} \int_a^b \mathbb{E}\left[\left\|\nabla \log \tilde{p}_t(\overrightarrow{X}_t)\right\|^2\right] \mathrm{d}t, \quad \frac{1}{b-a} \int_a^b \mathbb{E}\left[\left\|\nabla \log \tilde{p}_t^{(n)}(\overrightarrow{X}_t^{(n)})\right\|^2 \Big| \mathbf{Z}^{(n)}\right] \mathrm{d}t \leqslant K^2,$$

*with:*

$$K^2 := \frac{1}{b-a} \int_a^b \left(\mathrm{e}^{-2t} D^2 + d \frac{\mathrm{e}^{-4t}}{1 - \mathrm{e}^{-2t}}\right) \mathrm{d}t. \tag{13}$$

*Note that our proof actually yields a stronger result where $D$ is replaced by the order $2$ moment of $\mu$ (or $\widehat{\mu}_n$, respectively).*

*Proof.* By Fisher's identity [Efr11] and Jensen's inequality, we have that:

$$\begin{aligned}
\mathbb{E}\left[\left\|\nabla \log \tilde{p}_t(\overrightarrow{X}_t)\right\|^2\right] &= \mathbb{E}\left[\left\|\mathbb{E}\left[\nabla \log \tilde{p}_{t|0}(\overrightarrow{X}_t | \overrightarrow{X}_0) | \overrightarrow{X}_t\right]\right\|^2\right] \\
&\leqslant \mathbb{E}\left[\mathbb{E}\left[\left\|\nabla \log \tilde{p}_{t|0}(\overrightarrow{X}_t | \overrightarrow{X}_0)\right\|^2 | \overrightarrow{X}_t\right]\right] \\
&= \mathbb{E}\left[\left\|\nabla \log \tilde{p}_{t|0}(\overrightarrow{X}_t | \overrightarrow{X}_0)\right\|^2\right].
\end{aligned}$$

By the proof of Lemma 3.1, we have that:

$$\mathbb{E}\left[\left\|\nabla \log \tilde{p}_t(\overrightarrow{X}_t)\right\|^2\right] \leqslant \mathbb{E}\left[e^{-2t}\left(\left\|\overrightarrow{X}_0\right\|^2 + \frac{d}{e^{2t}-1}\right)\right].$$

We conclude by integrating over $[a,b]$. We obtain similarly the formula for $\overrightarrow{p}_t^{(n)}$. $\qquad\square$

**Lemma A.6.** *Let* $0 < a < b$ *and assume that* $\mu$ *has compact support bounded by* $D$. *Let* $Y$ *denote the random variable.*

$$Y := \frac{1}{b-a}\int_a^b \mathbb{E}\left[\left\|\nabla \log \tilde{p}_t(\overrightarrow{X}_t^Z)\right\|^2 \Big| Z\right]\mathrm{d}t, \tag{14}$$

*with* $Z \sim \mu$. *Then,* $Y$ *is sub-exponential with constant* $\|Y\|_{\psi_1} \lesssim K_1^2$, *with:*

$$K_1^2 \lesssim \frac{d}{1-e^{-2a}} + D^2.$$

*Proof.* Let $j \in \mathbb{N}^\star$ be an even integer. By Fisher's identity and the conditional Jensen's inequality, we have (as in the proof of the above lemma):

$$\mathbb{E}\left[\left\|\nabla \log \tilde{p}_t(\overrightarrow{X}_t)\right\|^j\right] = \mathbb{E}\left[\left\|\mathbb{E}\left[\nabla \log \tilde{p}_{t|0}(\overrightarrow{X}_t|\overrightarrow{X}_0)|\overrightarrow{X}_t\right]\right\|^j\right] \leqslant \mathbb{E}\left[\left\|-\frac{\overrightarrow{X}_t - e^{-t}\overrightarrow{X}_0}{1-e^{-2t}} + \overrightarrow{X}_t\right\|^j\right].$$

Let us denote $\Xi \sim N(0, I)$, independent of $\overrightarrow{X}_0$. We have

$$\mathbb{E}\left[\left\|\nabla \log \tilde{p}_t(\overrightarrow{X}_t)\right\|^j\right] \leqslant \mathbb{E}\left[\left\|e^{-t}\overrightarrow{X}_0 - \frac{e^{-2t}\sqrt{1-e^{-2t}}\Xi}{1-e^{-2t}}\right\|^j\right]$$

By Jensen's inequality, we have

$$\mathbb{E}\left[\left\|\nabla \log \tilde{p}_t(\overrightarrow{X}_t)\right\|^j\right] \leqslant 2^{j-1}e^{-jt}D^j + \frac{e^{-2jt}}{(1-e^{-2t})^{j/2}}\mathbb{E}\left[\|\Xi\|^j\right]$$

By applying again Jensen's inequality, we see that:

$$\mathbb{E}\left[\|Xi\|^j\right] = \mathbb{E}\left[\left(\sum_{k=1}^d \Xi_k^2\right)^{j/2}\right] \leqslant d^{j/2-1}\mathbb{E}\left[\Xi_1^j\right] \leqslant d^{j/2}C^j j^{j/2},$$

where the last inequality follows from the moments-based characterization of subgaussian distributions [Ver18, Proposition 2.5.2] and $C > 0$ is an absolute constant. We conclude that

$$\mathbb{E}\left[\left\|\nabla \log \tilde{p}_t(\overrightarrow{X}_t)\right\|^j\right] \leqslant d^{j/2}C^j j^{j/2} + 2^j D^j \leqslant j^{j/2}(K_1^2)^{j/2},$$

with

$$K_1^2 \lesssim \frac{d}{1-e^{-2t}} + D^2.$$

Hence, by Jensen's inequality we have:

$$\forall m \in \mathbb{N}^\star,\ \mathbb{E}\left[Y^m\right] \leqslant \frac{1}{b-a}\int_a^b \mathbb{E}\left[\left\|\nabla \log \tilde{p}_t(\overrightarrow{X}_t)\right\|^{2m}\right]\mathrm{d}t \leqslant \left(K_1^2\right)^m m^m.$$

By the moments-based characterization of sub-exponential random variables [Ver18, Proposition 2.7.1], we deduce that $Y$ is sub-exponential with sub-exponential norm (see [Ver18]) $\|Y\|_{\psi_1} \lesssim K_1^2$.

$\qquad\square$

The next lemma is a upper bound on the KL divergence between two distributions generated by Ornstein-Uhlenbeck processes at a time $t > 0$. In this form, this lemma is taken from Proposition 23 in [ADR24], which the authors notice can be traced back to [Vil09]. For the sake of completeness and for a minor correction of the constants, we provide a short proof of this known result.

**Remark A.1.** *Under our assumption on $\mu$, we can even show that the random variable $Y$ above is sub-gaussian, but this would give a worse final iterate than using the above lemma. Moreover, the above proof technique still works under weaker assumptions on the data distributions, such as a sub-gaussian or sub-exponential distribution.*

**Lemma A.7.** *Let $\mu$ and $\nu$ be two probability distributions on $\mathbb{R}^d$ and $0 < t_0 < T$. Let $p_t$ be the density of the OU process (2) initialized at $X_0 \sim \mu$ and let $q_t$ be the density of the OU process (2) initialized at $X_0 \sim \nu$. Then we have:*

$$\int_{t_0}^{T} \mathscr{I}(p_t | q_t) \mathrm{d}t \leqslant \mathrm{KL}(p_{t_0} | q_{t_0}) \leqslant \frac{1}{2(\mathrm{e}^{t_0} - 1)} \mathrm{W}_2(p_{t_0/2}, q_{t_0/2})^2.$$

*Proof.* For the purpose of this proof, let $L\phi := \Delta\phi - \langle x, \nabla\phi \rangle$ the generator of the semigroup $(P_t)_t$ of Equation (2). As $P_t$ and $L$ are self-adjoint with respect to $\gamma^d$, we have that $\partial_t \tilde{p}_t = L\tilde{p}_t$, where $\tilde{p}_t = p_t/\gamma^d$ is the density of $p_t$ with respect to $\gamma^d$. Then we easily see that $\partial_t \log \tilde{p}_t = L \log \tilde{p}_t + \|\nabla \log \tilde{p}_t\|^2$ (and similarly for $\tilde{q}_t$).

By exploiting the chain rule and the integration by parts for $L$ [CL17, Lemma 1.13], we have:

$$\frac{\mathrm{d}}{\mathrm{d}t} \mathrm{KL}(p_t | q_t) = \int \log\left(\frac{\tilde{p}_t}{\tilde{q}_t}\right) \tilde{p}_t \mathrm{d}\gamma^d$$

$$= \int \partial_t (\log \tilde{p}_t) \tilde{p}_t \mathrm{d}\gamma^d - \int \left(L \log \tilde{q}_t + \|\nabla \log \tilde{q}_t\|^2\right) \tilde{p}_t \mathrm{d}\gamma^d + \int \log\left(\frac{\tilde{p}_t}{\tilde{q}_t}\right) L\tilde{p}_t \mathrm{d}\gamma^d$$

$$= 0 + \int \langle \nabla \log \tilde{p}_t, \nabla \log \tilde{q}_t \rangle \tilde{p}_t \mathrm{d}\gamma^d - \int \|\nabla \log \tilde{q}_t\|^2 \tilde{p}_t \mathrm{d}\gamma^d$$

$$\quad - \int \|\nabla \log \tilde{p}_t\|^2 \tilde{p}_t \mathrm{d}\gamma^d + \int \langle \nabla \log \tilde{p}_t, \nabla \log \tilde{q}_t \rangle \tilde{p}_t \mathrm{d}\gamma^d$$

$$= -\int \|\nabla \log \tilde{p}_t - \nabla \log \tilde{q}_t\|^2 \tilde{p}_t \mathrm{d}\gamma^d$$

$$= -\mathscr{I}(p_t | q_t).$$

By integrating this relation and using the non-negativity of the KL divergence we obtain:

$$\int_{t_0}^{T} \mathscr{I}(p_t | q_t) \mathrm{d}t = \mathrm{KL}(p_{t_0} | q_{t_0}) - \mathrm{KL}(p_T | q_T) \leqslant \mathrm{KL}(p_{t_0} | q_{t_0}).$$

Let $X_0 \sim \mu$ and $Y_0 \sim \nu$. We now use the fact that $p_t$ is the probability density of $X_t := \mathrm{e}^{-t} X_0 + \sqrt{1 - \mathrm{e}^{-2t}} \mathrm{N}(0, I_d)$ and $q_t$ is the probability density of $Y_t := \mathrm{e}^{-t} Y_0 + \sqrt{1 - \mathrm{e}^{-2t}} \mathrm{N}(0, I_d)$. By the semigroup property and by joint convexity of the KL divergence [vH14] we have the known inequality (see also [NDHR21]):

$$\int_{t_0}^{T} \mathscr{I}(p_t | q_t) \mathrm{d}t \leqslant \frac{1}{2(1 - \mathrm{e}^{-t_0})} \mathrm{W}_2(\mathrm{Law}(\mathrm{e}^{-t_0/2} X_{t_0/2}), \mathrm{Law}(\mathrm{e}^{-t_0/2} Y_{t_0/2}))^2$$

$$= \frac{1}{2(\mathrm{e}^{t_0} - 1)} \mathrm{W}_2(q_{t_0/2}, p_{t_0/2})^2.$$

$\square$

**Proof of Proposition 3.1.**

*Proof.* By Lemma 3.1, we have that with probability at least $1 - \delta$ over $\mathbf{Z}^{(n)} = (Z_1, \ldots, Z_n) \sim \mu^{\otimes n}$:

$$\widehat{\Delta}_T^{(n)} \leqslant 4D^2 \sqrt{\frac{\log(1/\delta)}{2n}} \int \mathrm{e}^{-2t} \mathrm{d}\lambda(t) + \frac{4}{T} \sum_{k=0}^{N-1} h\left(\mathscr{I}(\overrightarrow{p}_{T-t_k} | \gamma^d) - \mathscr{I}(\overrightarrow{p}_{T-t_k}^{(n)} | \gamma^d)\right),$$

where we used the fact that $h_k = h$. We now apply Lemma A.4 to obtain that with probability at least $1 - \delta$ over $S \sim \mu^{\otimes n}$, we have

$$\widehat{\Delta}_T^{(n)} \leqslant 4D^2 \sqrt{\frac{\log(1/\delta)}{2n}} \int e^{-2t} \mathrm{d}\lambda(t) + \frac{4h}{T} \mathscr{I}(\overrightarrow{p}_{\tau_n} | \gamma^d) + \frac{4(T-h)}{T}(A_1 + A_2),$$

with, by adding and substracting:

$$A_1 := \frac{1}{T-h} \int_h^T \left( \mathbb{E}\left[ \left\| \nabla \log \tilde{p}_t(\overrightarrow{X}_t) \right\|^2 \right] - \frac{1}{n} \sum_{i=1}^n \mathbb{E}\left[ \left\| \nabla \log \tilde{p}_t(\overrightarrow{X}_t^{Z_i}) \right\|^2 | \mathbf{Z}^{(n)} \right] \right) \mathrm{d}t,$$

$$A_2 := \frac{1}{n} \sum_{i=1}^n \frac{1}{T-h} \int_h^T \left( \mathbb{E}\left[ \left\| \nabla \log \tilde{p}_t(\overrightarrow{X}_t^{Z_i}) \right\|^2 | \mathbf{Z}^{(n)} \right] - \mathbb{E}\left[ \left\| \nabla \log \tilde{p}_t^{(n)}(\overrightarrow{X}_t^{Z_i}) \right\|^2 | \mathbf{Z}^{(n)} \right] \right) \mathrm{d}t,$$

By Lemma A.6, we know that the random variable,

$$\frac{1}{T-h} \int_h^T \mathbb{E}\left[ \left\| \nabla \log \tilde{p}_t(\overrightarrow{X}_t^{Z}) \right\|^2 | Z \right] \mathrm{d}t,$$

is sub-exponential with constant $K_1^2$ with respect to $Z \sim \mu$, with:

$$K_1^2 \lesssim \frac{d}{1 - e^{-2h}} + D^2.$$

Recall that $\mathbf{Z}^{(n)} \sim \mu^{\otimes n}$. Hence, by Bernstein's inequality [Ver18, Theorem 2.8.1], we have that:

$$\mu^{\otimes n}(A_1 \geqslant \epsilon) \leqslant \exp\left( -cn \min\left( \frac{\epsilon^2}{K_1^4}, \frac{\epsilon}{K_1^2} \right) \right),$$

where $c > 0$ is an absolute constant. We deduce that with probability at least $1 - \zeta$ over $\mathbf{Z}^{(n)} \sim \mu^{\otimes n}$, we have:

$$A_1 \lesssim K_1^2 \left( \sqrt{\frac{\log(1/\zeta)}{n}} + \frac{\log(1/\zeta)}{n} \right).$$

In the following, we fix $\mathbf{Z}^{(n)}$, so that all expectations have to be understood as conditioned on $\mathbf{Z}^{(n)}$. We now turn our attention to $A_2$. We use the identity $\|a\|^2 - \|b\|^2 = \|a - b\|^2 + 2\langle b, a - b \rangle$, the Cauchy-Schwarz inequality, and Lemma A.5, to get,

$$A_2 \leqslant \frac{1}{n} \sum_{i=1}^n \frac{1}{T-h} \int_h^T \mathbb{E}\left[ \left\| \nabla \log \tilde{p}_t(\overrightarrow{X}_t^{Z_i}) - \nabla \log \tilde{p}_t^{(n)}(\overrightarrow{X}_t^{Z_i}) \right\|^2 \right] \mathrm{d}t$$

$$+ 2K' \sqrt{\frac{1}{n} \sum_{i=1}^n \frac{1}{T-h} \int_h^T \mathbb{E}\left[ \left\| \nabla \log \tilde{p}_t(\overrightarrow{X}_t^{Z_i}) - \nabla \log \tilde{p}_t^{(n)}(\overrightarrow{X}_t^{Z_i}) \right\|^2 \right] \mathrm{d}t}$$

$$= \frac{1}{T-h} \int_h^T \mathbb{E}\left[ \left\| \nabla \log \overrightarrow{p}_t(\overrightarrow{X}_t^{(n)}) - \nabla \log \overrightarrow{p}_t^{(n)}(\overrightarrow{X}_t^{(n)}) \right\|^2 \right] \mathrm{d}t$$

$$+ 2K' \sqrt{\frac{1}{T-h} \int_h^T \mathbb{E}\left[ \left\| \nabla \log \overrightarrow{p}_t(\overrightarrow{X}_t^{(n)}) - \nabla \log \overrightarrow{p}_t^{(n)}(\overrightarrow{X}_t^{(n)}) \right\|^2 \right] \mathrm{d}t}$$

$$= \frac{1}{T-h} \int_h^T \mathscr{I}(\overrightarrow{p}_t^{(n)} | \overrightarrow{p}_t) \mathrm{d}t + + 2K' \sqrt{\frac{1}{T-h} \int_h^T \mathscr{I}(\overrightarrow{p}_t^{(n)} | \overrightarrow{p}_t) \mathrm{d}t},$$

with:

$$K'^2 := \frac{1}{T-h} \int_h^T \left( e^{-2t} D^2 + d \frac{e^{-4t}}{1 - e^{-2t}} \right) \mathrm{d}t \tag{15}$$

$$\lesssim \frac{D^2}{T-h} + \frac{d}{T-h} \log(T/h) + \frac{hd}{T-h}. \tag{16}$$

By Lemma A.7 we obtain:

$$\frac{T-h}{T}A_2 \lesssim \frac{1}{T}\left(\mathcal{K}_h + 2K_2\sqrt{\mathcal{K}_h}\right),$$

with $K_2^2 := D^2 + d\log(T/h) + hd$ and $\mathcal{K}_h := \mathrm{KL}\left(\overrightarrow{p}_h^{(n)}|\overrightarrow{p}_h\right)$. From the second part of Lemma A.7, we also deduce that:

$$\frac{T-h}{T}A_2 \lesssim \frac{W^2}{2T\left(\mathrm{e}^h-1\right)} + 2K\sqrt{\frac{W^2}{2T\left(\mathrm{e}^h-1\right)}} \leqslant \frac{W^2}{2Th} + 2\frac{K_2}{T}\sqrt{\frac{W^2}{2h}},$$

with $W := W_2(\overrightarrow{p}_{\frac{h}{2}}, \overrightarrow{p}_{\frac{h}{2}}^{(n)})$.

We conclude by a union bound that with probability at least $1-2\delta$ over $\mathbf{Z}^{(n)} \sim \mu^{\otimes n}$, we have:

$$\widehat{\Delta}_T^{(n)} \lesssim \left(D^2 + K_1^2\right)\sqrt{\frac{\log(1/\delta)}{2n}} + \frac{h}{T}\mathscr{I}(\mu|\gamma^d) + K_1^2\frac{\log(1/\delta)}{n} + \frac{W^2 + K_2\sqrt{h}W}{Th}.$$

with:

$$K_2^2 := D^2 + 2d\log(T/h) + hd, \quad K_1^2 := \frac{d}{1-\mathrm{e}^{-2h}} + D^2, \quad W := W_2\left(\overrightarrow{p}_{h/2}, \overrightarrow{p}_{h/2}^{(n)}\right).$$

$\square$

# B  Omitted proofs and additional details on the generalization bounds

## B.1  Noisy SGD

Consider the SGLD recursion, with $a > 0$ a regularization parameter.

$$\theta_{k+1} = (1-a\eta_k)\theta_k - \eta_k\widehat{g}_k + \sqrt{\frac{2\eta_k}{\beta}}\xi_k, \quad \theta_0 \sim \nu_0, \tag{17}$$

with $\widehat{g}_k$ an unbiased estimate of the gradient of the empirical risk, and $\xi_k \sim \gamma^d = \mathrm{N}(0, \mathrm{I}_d)$ independent of $\widehat{g}_k$. We denote by $p_k$ the distribution of $\widehat{g}_k$.

The proof presented here is an adaptation of [MWZZ18] and is also inspired by the so-called half step technique used in [DC25]. The main difference with [MWZZ18] is to remove the need for a Lipschitz continuity assumption of the loss by using a recently proposed PAC-Bayesian bound adapted to sub-Gaussian losses [DS24]. The proof is based on classical arguments in the noisy SGD literature, but we present it for the sake of completeness.

**Proof of Theorem 4.1**

*Proof.* We consider a "prior" stochastic process defined as $X_{k+1} = (1-a\eta_k)X_k + \sqrt{2\eta_k\beta^{-1}}\xi_k$ with $X_0 \sim \nu_0 = \mathrm{N}(0, \sigma_0^2)$. Then it is clear that $X_k \sim \pi_k := \mathrm{N}(0, \sigma_k^2 I_d)$ with $\forall k \in \mathbb{N}$, $\sigma_{k+1}^2 = (1-a\eta_k)^2\sigma_k^2 + 2\eta_k\beta^{-1}$. Then we can see by recursion that $\forall k \in \mathbb{N}$, $\sigma_k\sqrt{\beta a} \leqslant \sqrt{2}$.

Thanks to the subgaussian assumption, we can apply Theorem 2.1 of [DS24] to obtain that with probability at least $1-\delta$ over $\mathbf{Z}^{(n)} \sim \mu^{\otimes n}$, we have:

$$\mathbb{E}_{\theta\sim p_N}\left[\mathscr{G}_\lambda(\mathbf{Z}^{(n)}, \theta)\right] \leqslant 2\Sigma\sqrt{\frac{\mathrm{KL}(p_N|\pi_N) + \log(3/\delta)}{n}}. \tag{18}$$

Now let us fix some $k \in \mathbb{N}$ and introduce $p_k(t)$ and $\pi_k(t)$ be the distribution of $\theta_k(t) := (1-a\eta_k)\theta_k - \eta_k\widehat{g}_k + \sqrt{2t}\xi_k$ and $X_k(t) := (1-a\eta_k)X_k + \sqrt{2t}\xi_k$, respectively, for $t \in [0, \eta_k\beta^{-1}]$. Let $\sigma_k(t)^2$ be the variance $\pi_k(t)$. Then we have the following identity, which is a generalization of De-Bruijn's identity (see for instance [CCSW22] or our proof of Lemma A.7), for $t > 0$:

$$\frac{\mathrm{d}}{\mathrm{d}t}\mathrm{KL}(p_k(t)|\pi_k(t)) = -\mathscr{I}(p_k(t)|\pi_k(t))$$

By the logarithmic Sobolev inequality of $\pi_k(t)$, we have:

$$\frac{\mathrm{d}}{\mathrm{d}t}\mathrm{KL}(p_k(t)|\pi_k(t)) \leqslant -\frac{2}{\sigma_k(t)^2}\mathrm{KL}(p_k(t)|\pi_k(t)) \leqslant -\beta a\mathrm{KL}(p_k(t)|\pi_k(t)),$$

where the last line follows from $\sigma_k(t)^2 \leqslant 2/(\beta a)$. By integrating we find that:

$$\mathrm{KL}(p_{k+1}|\pi_{k+1}) \leqslant \mathrm{e}^{-\frac{\eta_k a}{2}}\mathrm{KL}(p_k(\eta_k\beta^{-1}/2)|\pi_k(\eta_k\beta^{-1}/2)).$$

We know apply the data processing inequality and the chain rule for KL divergence to get:

$$\mathrm{KL}(p_k(\eta_k\beta^{-1}/2)|\pi_k(\eta_k\beta^{-1}/2)) \leqslant \mathrm{KL}(\mathrm{Law}(\theta_k,\theta_{k+1/2})\,\mathrm{Law}(X_k,X_{k+1/2}))$$
$$\leqslant \mathrm{KL}(p_k|\pi_k) + \mathbb{E}_{x\sim p_k}\left[\mathrm{KL}(\mathrm{Law}(\theta_{k+1/2}|\theta_k=x)|\mathrm{Law}(X_{k+1/2}|X_k=x))\right].$$

By joint convexity of KL divergence (see also [NDHR21]), we now have:

$$\mathbb{E}_{x\sim p_k}\left[\mathrm{KL}(\mathrm{Law}(\theta_{k+1/2}|\theta_k=x)|\mathrm{Law}(X_{k+1/2}|X_k=x))\right] \leqslant \frac{\beta}{2\eta_k}\mathbb{E}\left[\|\eta_k\widehat{g_k}\|^2\right] = \frac{\beta}{2}\eta_k\mathbb{E}\left[\|\widehat{g_k}\|^2\right].$$

Therefore:

$$\mathrm{KL}(p_{k+1}|\pi_{k+1}) \leqslant \mathrm{e}^{-\frac{\eta_k a}{2}}\left(\mathrm{KL}(p_k|\pi_k) + \frac{\beta}{2}\eta_k\mathbb{E}\left[\|\widehat{g_k}\|^2\right]\right).$$

This implies that:

$$\mathrm{KL}(p_N|\pi_N) \leqslant \frac{\beta}{2}\sum_{k=0}^{N-1}\eta_k\mathrm{e}^{-\frac{a}{2}(S_N-S_k)}\mathbb{E}\left[\|\widehat{g_k}\|^2\right].$$

with $S_k := \sum_{j=0}^{k-1}\eta_j$. This implies the desired result by using Equation (18).

**Case where** $a = 0$. In that case, we apply the data processing inequality and the chain rule for KL divergence to obtain that:

$$\mathrm{KL}(p_N|\pi_N) \leqslant \mathrm{KL}(\mathrm{Law}(\theta_0,\ldots,\theta_N)|\mathrm{Law}(X_0,\ldots,X_N))$$
$$\leqslant \sum_{k=0}^{N-1}\mathbb{E}_{x\sim p_k}\left[\mathrm{KL}(\mathrm{Law}(\theta_{k+1}|\theta_k=x)|\mathrm{Law}(X_{k+1}|X_k=x))\right]$$
$$\leqslant \sum_{k=0}^{N-1}\frac{\beta}{4\eta_k}\mathbb{E}_{x\sim p_k}\left[\|\eta_k\widehat{g_k}\|^2\right],$$

where the last inequality follows again from the joint convexity of the KL divergence. This leads to the desired result. $\qquad\square$

## B.2 Background and additional results on topological complexities

In this section, we present some technical background on the topological complexities considered in Section 4.3 and also provide some additional topological generalization bounds for the score generalization gap.

### B.2.1 Information-theoretic terms

In this subsection, we quickly define the information-theoretic terms appearing in the topological bounds presented in Section 4.3. These terms come from the PAC-Bayesian theory on random sets introduced in [DS24]. While several choices are possible, we focus in our work on the total mutual information $I_\infty$, which has the advantage of yielding high-probability bounds. Note however that it could be replaced with the usual mutual information in the case of expected bounds fro isntance.

**Definition 1** (Total mutual information). *Consider two random variables $X$ and $Y$ on an arbitrary probability space and with values in measurable spaces $(\Omega_X, \mathcal{F}_X)$. The total mutual information between $X$ and $Y$ is defined as:*

$$I_\infty(X,Y) := \sup_{B\in\mathcal{F}_X\otimes\mathcal{F}_Y}\left(\frac{\mathbb{P}_{X,Y}(B)}{\mathbb{P}_X\otimes\mathbb{P}_Y}\right).$$

In the context of learning theory, this quantity has been used in several studies [HcSKM22, DDS23].

### B.2.2 Definition of weighted lifetime sums

In this section, we quickly provide additional technical background on the topological complexities mentioned in Section 4.

Consider a finite set $\mathcal{W}$ (which is Section 4 we take to be the trajectory $\mathcal{W}^{(n)}$) and a pseudometric $\rho$ on $\mathcal{W}$. There exist two equivalent definitions of the weighted lifetime sums used in Section 4: one using the notion of *persistent homology* [BCY18] and a definition based on minimum spanning trees [Sch20], which we present here for the sake of simplicity. See [ADS$^+$24, BLGcS21, DV23] for additional details and connections to learning theory. We first introduce the following definition.

**Definition 2** (Spanning tree). *A tree $\mathcal{T}$ over $\mathcal{W}$ is a connected acyclic (undirected) graph over $\mathcal{W}$. We represent it by a set of edges, where each edge is denoted $\{a, b\} \in \mathcal{T}$. The cost of an edge $\{a, b\}$ is set to be $\rho(a, b)$ and the cost of $\mathcal{T}$ is defined as:*

$$\mathscr{C}(\mathcal{T}) := \sum_{\{a,b\} \in \mathcal{T}} \rho(a, b).$$

We can now define the weighted-lifetime sums (of order 1).

**Definition 3** (Weighted lifetime sums). *The (1-)weighted lifetime sum of $\mathcal{W}$ is the cost $\mathscr{C}(\mathcal{T})$ of a spanning tree of $\mathcal{W}$ with minimal cost.*

In the context of generalization bounds, several choices are possible for the choice of the pseudometric $\rho$ [ADS$^+$24, Section 3.1]. In our paper, we focus on the particular choice of the so-called *data-dependent pseudometric* [DDS23], which gives the most promising empirical results in existing works. Given a dataset $\mathbf{Z}^{(n)} = (Z_1, \ldots, Z_n) \sim \mu^{\otimes n}$, we define the vectors $\ell_{\mathbf{Z}^{(n)}}(w) := (\ell_\lambda(w, Z_i)_{1 \leqslant i \leqslant n}) \in \mathbb{R}^n$. The data-dependent pseudometric is then defined as:

$$\rho_{\mathbf{Z}^{(n)}}(w, w') := \frac{1}{n} \sum_{i=1}^{n} \|\ell_{\mathbf{Z}^{(n)}}(w) - \ell_{\mathbf{Z}^{(n)}}(w')\|, \tag{19}$$

where $\|\cdot\|$ is a norm on $\mathbb{R}^n$, which can be the $\ell^1$ or $\ell^2$ norm. The $\ell^1$ is mostly used in [ADS$^+$24] and it is also our choice in our work.

The quantity $E_1(\mathcal{W}^{(n)})$ appearing in Theorem 4.2 is defined as the weighted lifetime sum of $\mathcal{W}^{(n)}$ for the pseudometric $\rho_{\mathbf{Z}^{(n)}}$.

### B.3 Positive magnitude bounds

We start by defining the notion of positive magnitude. Magnitude was initially introduced in [Lei13] and positive magnitude is a variant introduced by [ADS$^+$24], which is more suited to learning theory. While a more general definition is possible, we focus here on the case of finite sets, as the discrete-time stochastic optimizers considered in our study generate finite trajectories. In the following, let $(\mathcal{W}, \rho)$ be a finite metric space as in the above subsection. Given a positive scale parameter $r > 0$, we say that $(\lambda \dot{\mathcal{W}})$ has magnitude [Mec15] is there exists a vector $\beta : \mathcal{W} \to \mathbb{R}$ (called a weighting) such that:

$$\forall a \in \mathcal{W}, \ \sum_{b \in \mathcal{W}} e^{-r\rho(a,b)} \beta(b) = 1.$$

This has been shown to be satisfied for the metric spaces considered in our study [Mec15, ADS$^+$24]. The positive magnitude is then defined as:

$$\mathrm{PMag}(r \cdot \mathcal{W}) := \sum_{a \in \mathcal{W}} \beta(a)_+,$$

where $\beta_+$ denotes the positive part of $\beta$.

In all our work, we take $\rho$ to be the data-dependent pseudometric defined in Equation (19).

Applying [ADS$^+$24, Theorem 3.5], we obtain the following bound on the score generalization gap.

**Theorem B.1.** *Assume that the loss $\ell_\lambda(\theta, z)$ is uniformly bounded by $B > 0$ and that we have a Fisher information $\mathscr{I}(\mu|\gamma^d) < +\infty$ and use constant step size $h_k = h$. Then, with probability at least $1 - \delta$, we have for all $\theta \in \mathcal{W}^{(n)}$ and all $r > 0$ that:*

$$\mathscr{G}_\lambda(\mathbf{Z}^{(n)}, \hat{\theta}^{(n)}) \leqslant \frac{2}{r} \log \mathrm{PMag}(Lr \cdot \mathcal{W}^{(n)}) + r\frac{B^2}{n} + 3B\sqrt{\frac{I + \log\left(\frac{1}{\delta}\right)}{n}},$$

*where $I := I_\infty(\mathcal{W}^{(n)}, \mathbf{Z}^{(n)})$ is a total mutual information term and $K_n := 4\sqrt{n}/B$.*

## C  Experimental Details

In this section we provide full details regarding the experimental setup. All experiments are implemented using PyTorch.

**Forward process.** We use the Ornstein–Uhlenbeck (OU) process for the forward diffusion, also known as the variance-preserving process [SSDK$^+$21]. To characterize the conditional law $p_{t|0}$ of $\overrightarrow{X}_t$ given $\overrightarrow{X}_0$, we define $\alpha : t \mapsto \exp(-2t)$. The process admits the following reparameterization:

$$\overrightarrow{X}_t \stackrel{\mathrm{d}}{=} \sqrt{\alpha(t)}\overrightarrow{X}_0 + \sqrt{1 - \alpha(t)}G, \quad \text{where } G \sim \mathrm{N}(0, \mathrm{I}_d). \tag{20}$$

We adopt the cosine schedule introduced in [DN21] for both training and sampling. Specifically, we construct a discretization $\{\bar{t}_i\}_{i=1}^N$ of $[0, 1]$ with $N$ equally spaced points, and define:

$$\bar{\alpha}_{\bar{t}_i} = \frac{f(\bar{t}_i)}{f(0)}, \quad \text{with} \quad f(t) = \cos\left(\frac{t + s}{1 + s} \cdot \frac{\pi}{2}\right), \quad s = 0.008.$$

To ensure numerical stability, we truncate the schedule to $[0, 1 - \zeta]$, where $\zeta$ is chosen such that $1 - \bar{\alpha}(\bar{t}_N)/\bar{\alpha}(\bar{t}_{N-1}) \leqslant 0.999$. This prevents divergence of the final time $T$ and step size $h_N$, and ensures that bounds remain finite, as recommended in [DN21]. The final time grid $\{t_i\}_{i=1}^N$ is then obtained by solving $\alpha(t_i) = \bar{\alpha}(\bar{t}_i)$ for each $i$.

**Training and $\epsilon$-parameterization.** We adopt the $\epsilon$-parameterization introduced in [HJA20b], widely used in subsequent works [DN21, HS21, SH22, KAAL22b, EKB$^+$24]. This choice is motivated by practical considerations: improved numerical stability and faster convergence. The standard DSM loss exhibits high variance, resulting in noisier gradients and losses. We do not explore alternative approaches such as v-prediction [SH22].

The score model is parameterized as:

$$s_\theta(t, x) = \frac{-2\epsilon_\theta(t, x)}{\sqrt{1 - \alpha(t)}}, \tag{21}$$

which is motivated by the identity $2\nabla \log \overrightarrow{p}_{t|0}(x|x_0) = -2(x - \sqrt{\alpha(t)}x_0)/(1 - \alpha(t))$, together with the expression of the noise term in (20).

We define a *simulation-free* process $(\tilde{X}_t)_{0\leqslant t\leqslant T}$ via $\tilde{X}_t^Z = \sqrt{\alpha(t)}Z + \sqrt{1 - \alpha(t)}G_t$, where $(G_t)_{0\leqslant t\leqslant T}$ are i.i.d. samples from $\mathrm{N}(0, \mathrm{I}_d)$. The training objective is the $\epsilon$-loss:

$$\widetilde{\mathcal{L}}_{\epsilon-\mathrm{DSM}}(\theta) := \frac{1}{n}\sum_{i=1}^n \int_0^T \mathbb{E}\left[\left\|\epsilon_\theta(t, \tilde{X}_t^{Z_i}) - G_t\right\|^2 \mid Z_i\right]\nu(\mathrm{d}t), \quad \text{where } \nu = \mathrm{Unif}(\{t_i\}_{i=1}^N). \tag{22}$$

For reference, the original DSM loss integrates with respect to $\lambda = T^{-1}\sum_{k=1}^N h_{N-k}\delta_{t_k}$, as in Section 2. Notably, at a fixed timestep $t$, the $\epsilon$-loss equals the DSM loss up to a multiplicative factor $w(t) = 1/(1-\alpha(t))$ [HJA20b], which diverges near $t = 0$, causing instability and increased variance. Thus, minimizing the $\epsilon$-loss effectively corresponds to minimizing the DSM loss.

During training, for each datapoint in a batch, we sample a single timestep $t \sim \nu$ and a single Gaussian variable $G_t$ to evaluate the stochastic objective in (22). For evaluation on the train and test datasets, to reduce the variance of the estimated losses, we sample 10 independent timesteps and 10 corresponding noise terms per datapoint, and average the resulting losses.

**Generative process.** To sample from the trained model, we use the Euler–exponential integrator described in (6).

**Compute resources.** Experiments are conducted using 8 NVIDIA A100 GPUs. A training run of 100,000 steps on `MNIST` takes approximately 3 hours on a single GPU. Sampling 2,000 images with 500 reverse steps takes approximately 10 minutes on a single GPU. The VRAM consumption typically varies between 4-20GB depending on the chosen hyper-parameter configuration. The total computational budget for this work amounts to approximately 1,250 GPU hours.

## C.1  Low-dimensional dataset

We provide additional details on our low-dimensional experiments used to validate our SGLD bounds in Section 4.2.

**Mixture of Gaussian dataset.** The dataset consists of a mixture of nine Gaussian distributions in $\mathbb{R}^4$:

$$\sum_{i=1}^{9} w_i \cdot \mathcal{N}(\mu_i, \sigma^2 \mathrm{I}_4), \tag{23}$$

where the weights are $(w_i)_{i=1}^{9} = (0.01,\ 0.1,\ 0.3,\ 0.2,\ 0.02,\ 0.15,\ 0.02,\ 0.15,\ 0.05)$, and we fix $\sigma = 0.05$. The component means $(\mu_i)_{i=1}^{9}$ are sampled once uniformly at random from $[-1,1]^4$. We observed no significant difference in generative performance across seeds or across handcrafted choices, like arranging means in a grid-like pattern.

**Model architecture.** We use a neural network consisting of three time-conditioned MLP blocks with skip connections. Each block consists of two hidden layers of width 32. The input timestep $t$ is first processed through two fully connected layers of size $32 \times 32$, and then passed to each MLP block via an additional $32 \times 32$ transformation before being added (element-wise) to the intermediate representation in the block.

**SGLD generalization bounds.** The optimization is carried out using Stochastic Gradient Langevin Dynamics (SGLD) with no momentum or weight decay, using the `torch-sgld` package. Models are trained for 100,000 steps using $N = 1000$ forward timesteps. We vary the inverse temperature parameter $\beta \in \{10^4, 10^6, 10^{10}\}$ (i.e., $T = 1/\beta$). We also sweep over the step size, batch size, and dataset size, with batch size equal to the number of samples. Specifically:

- Step size $\in \{2\text{e}{-4},\ 5\text{e}{-4},\ 1\text{e}{-3},\ 2\text{e}{-3},\ 5\text{e}{-3}\}$,
- Total number of samples($=$ batch size) $\in \{512,\ 1024,\ 2048,\ 4096,\ 8192\}$.

For each hyper-parameter configuration, we run 10 experiments with different random seeds.

**Evaluation.** The number of steps at inference is fixed to $N = 100$, where all models are observed to perform optimally. To evaluate generative quality, we compute the Wasserstein-2 ($\mathcal{W}_2$) distance between the generated and target data distributions. The squared $\mathcal{W}_2$ distance between two distributions $\mu$ and $\nu$ over $\mathbb{R}^d$ is given by:

$$\mathcal{W}_2^2(\mu, \nu) = \inf_{\gamma \in \mathcal{M}(\mu,\nu)} \int \|x - y\|_2^2 \gamma(\mathrm{d}x, \mathrm{d}y),$$

where $\mathcal{M}(\mu, \nu)$ denotes the set of all couplings between $\mu$ and $\nu$. In our case, we compute the empirical $\mathcal{W}_2^2$ distance between 25,000 generated samples and 25,000 real samples using the `pyemd` package [Las17], with default settings.

## C.2  Image data

We consider three image datasets to validate our bounds in Section 4.3: MNIST, the `butterflies` dataset [WME09], and the `flowers17` dataset [NZ06], simply referred to as `flowers`.

**Model architecture.** Our implementation relies on the U-Net architecture from [DN21], available at https://github.com/openai/improved-diffusion, and with configurations described in Table 1. The activation function is SiLU, and self-attention is applied at the specified resolutions. The diffusion time $t$ is rescaled to lie in $(0, 1)$ and encoded via the Transformer sinusoidal position embedding [VSP$^+$17].

| Configuration | MNIST | butterflies | flowers |
|---|---|---|---|
| Input dimension | $1 \times 32 \times 32$ | $3 \times 64 \times 64$ | $3 \times 64 \times 64$ |
| attn_resolutions | [2, 4] | [4, 8, 16] | [4, 8, 16] |
| channel_mult | [1, 2, 2, 2] | [1, 2, 2, 2, 4] | [1, 2, 4, 4] |
| model_channels | 32 | 128 | 64 |
| num_res_blocks | 2 | 2 | 2 |
| num_heads | 4 | 4 | 4 |
| dropout | 0.0 | 0.0 | 0.0 |

Table 1: U-Net architecture configurations for each image dataset.

**MNIST.** Images are resized from $28 \times 28$ to $32 \times 32$. Each model is trained for 500,000 steps.

**Butterflies and Flowers.** All images are resized to $3 \times 64 \times 64$. The butterflies dataset consists of 702 training images and 130 test images; the flowers dataset consists of 1,020 training images (combining train and validation sets), and 340 test images. Each model is trained for 200,000 steps.

**Evaluation.** We use $N = 500$ steps during sampling. We evaluate generative performance using the Fréchet Inception Distance (FID) [HRU$^+$17]. For MNIST, we compute the FID using 2,048 generated images compared against 2,048 real images, once using the training set (train FID) and once using the test set (test FID). We refer to the latter simply as "FID" throughout our experiments. For the butterflies and flowers datasets, we match the number of generated images to the number of real images in the train/test splits.

**ADAM generalization bounds.** We use the Adam optimizer [KB17] for training. For each model, the time discretization during training corresponds to $N = 4000$. We vary the learning rate and batch size across:

- Learning rates: $\{5e-6, 1e-5, 2e-5, 1e-4, 2e-4\}$,

- Batch sizes: $\{4, 16, 64, 128\}$.

To study the generalization properties of the trained score models, we compute topological bounds by monitoring their behavior during optimization over a fixed subset of the training data of size $\min(\text{dataset size}, 3000)$. Moreover, we also sample and fix a single noise term and a single timestep per datapoint using the same random seed, computing loss terms on the same datapoint, timestep, noise term triplets across all configurations, obtaining comparable trajectories. For each iteration, we evaluate the per-subset $\epsilon$-loss and score loss, allowing us to track the evolution of local training dynamics. This procedure is repeated for 5,000 optimization steps over the train set starting from each fully trained model, where each train loss computation and optimization step is done as described in the introductory paragraphs of Appendix C.

The resulting trajectories are then analyzed and topological complexities (weighted lifetime sums and positive magnitude) are then computed using the procedures described in Appendix B.2. As it is the case in [ADS$^+$24], we consider the $\alpha$-weighted lifetime sums with $\alpha = 1$. Regarding positive magnitude (denoted $\text{PMag}(\lambda \cdot \mathcal{W}^{(n)})$), we make two choices, as briefly discussed in Section 4.3:

- The first choice is to take $\lambda = \sqrt{n}$, which is the theoretical value suggested in [ADS$^+$24].

- It was also argued by these authors that small values of the scale parameter can yield good correlation with the generalization error. As they do in their study, we also report experiments using the value $r = 10^{-2}$.

For the butterflies and flowers dataset, we used the whole training trajectory to evaluate the data-dependent pseudometric (19). For the MNIST dataset, in order to reduce the storage and computational costs of this eperiment, we preselected a subset of size 3000 of the training set and used it to estimate (19). This procedure is standard in the literature and experiments have shown that it accurately estimates the topological complexity [DDS23, ADS$^+$24].

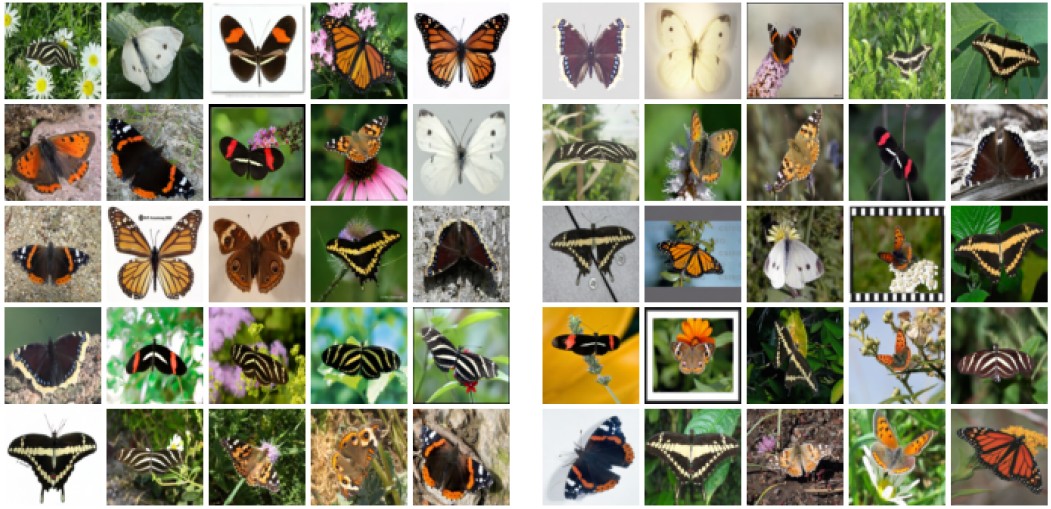

(a) ADAM, $\eta = 10^{-4}$, batch size $= 16$          (b) ADAM, $\eta = 10^{-5}$, batch size $= 128$

Figure 4: `butterflies` samples after 200k training steps, 500 sampling steps.

Table 2: ADAM on `flowers`. Test/Train Denoising Score-Matching loss $\mathcal{R}^{(\lambda)}(\theta), \widehat{\mathcal{R}}^{(\lambda)}_{\mathbf{Z}^{(n)}}(\theta)$ vs. complexity indicators and hyperparameters.

| | $\mathcal{R}^{(\lambda)}(\theta)$ | $\widehat{\mathcal{R}}^{(\lambda)}_{\mathbf{Z}^{(n)}}(\theta)$ | $b \times \langle\|\widehat{g}\|^2\rangle$ | $E^1(\mathcal{W}^{(n)})$ | $\mathrm{PMag}(10^{-2}\mathcal{W}^{(n)})$ | $\mathrm{PMag}(\sqrt{n}\mathcal{W}^{(n)})$ | $\eta$ | $b$ |
|---|---|---|---|---|---|---|---|---|
| 1 | 0.0773 | 0.0296 | 0.042 | 480.8 | 27.29 | 4396 | 0.000100 | 4.00 |
| 2 | 0.0811 | 0.0128 | 0.087 | 564.7 | 42.88 | 4605 | 0.000100 | 16.00 |
| 3 | 0.0842 | 0.0262 | 0.082 | 453.8 | 28.18 | 4290 | 0.000050 | 4.00 |
| 4 | 0.0851 | 0.0126 | 0.207 | 558.5 | 41.98 | 4573 | 0.000050 | 16.00 |
| 5 | 0.0859 | 0.0073 | 0.526 | 615.8 | 52.18 | 4698 | 0.000050 | 64.00 |
| 6 | 0.0865 | 0.0074 | 0.242 | 628.1 | 52.91 | 4712 | 0.000100 | 64.00 |
| 7 | 0.0894 | 0.0129 | 0.959 | 591.4 | 41.97 | 4625 | 0.000020 | 16.00 |
| 8 | 0.0948 | 0.0084 | 2.470 | 746.6 | 51.74 | 4835 | 0.000020 | 64.00 |
| 9 | 0.0959 | 0.0066 | 0.397 | 637.3 | 58.81 | 4731 | 0.000100 | 128.0 |
| 10 | 0.0972 | 0.0264 | 0.337 | 460.4 | 27.67 | 4302 | 0.000020 | 4.00 |
| 11 | 0.0998 | 0.0266 | 1.270 | 497.6 | 27.42 | 4391 | 0.000010 | 4.00 |
| 12 | 0.1023 | 0.0065 | 0.932 | 667.2 | 57.14 | 4757 | 0.000050 | 128.0 |
| 13 | 0.1071 | 0.0146 | 3.690 | 683.2 | 40.22 | 4735 | 0.000010 | 16.00 |
| 14 | 0.1116 | 0.0075 | 4.220 | 840.7 | 57.32 | 4894 | 0.000020 | 128.0 |
| 15 | 0.1134 | 0.0094 | 9.890 | 831.0 | 50.90 | 4879 | 0.000010 | 64.00 |
| 16 | 0.1362 | 0.0085 | 19.380 | 1005.0 | 55.01 | 4942 | 0.000010 | 128.0 |

### C.3  Additional results and remarks

In this section, we provide additional experiments to complement the discussion in Section 4. We also make several remarks on possible extensions to other transport-based generative models.

**Experiments on the `flowers` dataset.**   Table 2 reports Train/Test DSM together with $b \times \langle\|\widehat{g}\|^2\rangle$, $E^1(\mathcal{W}^{(n)})$, and $\mathrm{PMag}(\cdot)$ across learning rates and batch sizes for ADAM on `flowers`, complementing Fig. 3.

**Experiments on the `butterfly` dataset.**   We provide some generated image grids for the `butterfly` dataset in Figure 4 for visual comparison.

**Experiments on the `MNIST` dataset.**   We also computed the complexity presented in Sections 4.2 and 4.3 for the `MNIST` dataset. The results are reported in Figures 5 and 6.

Regarding the gradient norms-based bound studied in Section 4.2, we observe a quite satisfying correlation with the generalization error, apart from some points in the figure. We see in Figure 6 that it is related to the value of the train loss, suggesting that the models need to reach a certain threshold of convergence for our gradient-based complexity to be more relevant to understand generalization.

The weighted lifetime sums $E^1$ yield a slightly more contrasted result. As we observed for the gradient bound, this behavior seems to be connected to the train loss and, hence, the convergence of the model (see Figure 6). These observations are coherent with existing works on topological generalization bounds suggesting that they characterize geometric properties of local minima and, thus, the experiments require the models to reach such a local minimum [BLGcS21, DDS23, ADS⁺24]. As we are the first to evaluate these quantities for diffusion models, we observe that these conclusion seem to hold also in this case.

Regarding the positive magnitude, the observed correlation for $\mathrm{PMag}(10^{-2} \cdot \mathcal{W}^{(n)})$ is satisfying, even though the lack of convergence of certain experiments might be affecting the result. An interesting behavior can be observed for $\mathrm{PMag}(\sqrt{n} \cdot \mathcal{W}^{(n)})$, where most points attain the maximum value of $5 \cdot 10^3$. This is a known phenomenon that can happen with positive magnitude when the scale is not adapted and it is the reason that prompted the authors of [ADS⁺24] to introduce $\mathrm{PMag}(10^{-2} \cdot \mathcal{W}^{(n)})$. Overall, these experiments suggest that $\mathrm{PMag}(10^{-2} \cdot \mathcal{W}^{(n)})$ might be the best complexity metric for more complex datasets, which is in line with the results of [ADS⁺24, Table 1].

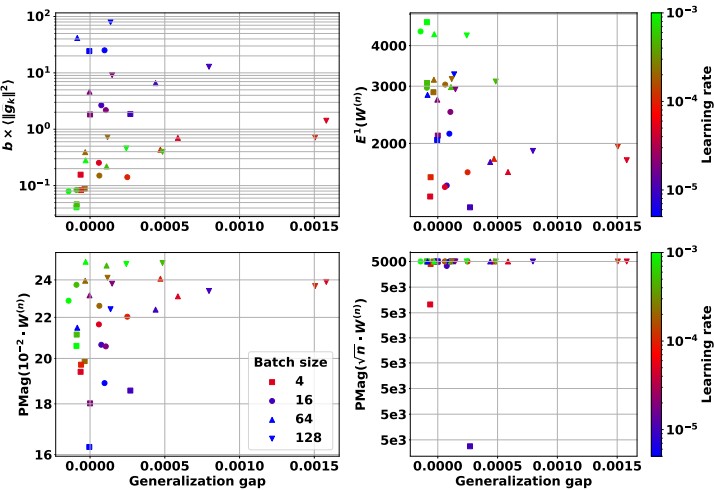

Figure 5: ADAM optimizer on `MNIST` dataset. Score generalization gap vs. several complexity metrics: $b\langle \|\widehat{g}_k\|^2\rangle$ *(top left)*, $E^1(\mathcal{W}^{(n)})$ *(top right)*, $\mathrm{PMag}(10^{-2} \cdot \mathcal{W}^{(n)})$ *(bottom left)* and $\mathrm{PMag}(\sqrt{n} \cdot \mathcal{W}^{(n)})$ *(bottom right)*.

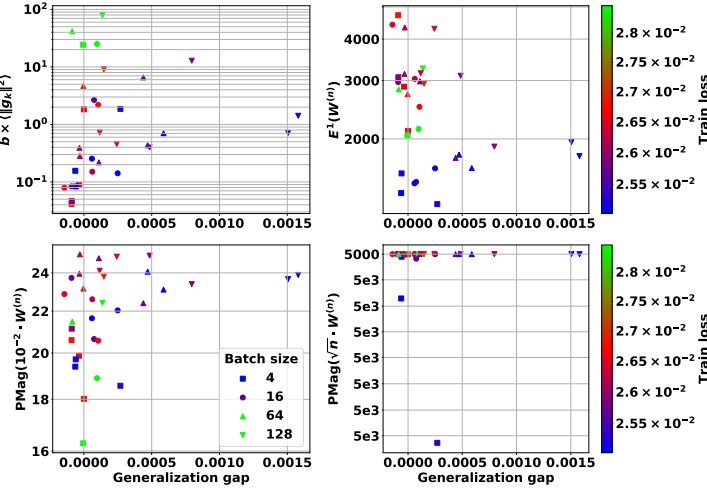

Figure 6: Same experiment as Figure 5, except that the color bar represents the value of the train loss instrad of the learning rate.

**Possible extensions beyond diffusion.** We expect that similar generalization phenomena may be experimentally observed in other classes of generative models. In particular, extensions to more general continuous state spaces could be considered, such as the setting of bridge matching models [Pel23, LGL22, LCBH+23], which admit a more flexible structure between base/target distributions. Discrete state spaces could be an interesting avenue of research too [AJH+23, CBB+22, LME24], especially in structured settings like the hypercube [PSO+25, BS25], where sharper convergence bounds have been obtained. Another direction could be generative models built upon alternative noise distributions, such as heavy-tailed distributions, which exhibit behaviors like robustness to mode collapse which could be tied to generalization abilities, see [YPKL23] for the continuous time regime, or [SSD25, DSL22] for a discrete time regime. This would require advanced concentration tools, going beyond classical assumptions (sub-Gaussian etc.) as distributions are unbounded or lack finite variance, complicating the analysis of both $\Delta_{\mathrm{s}}^{(n)}$ and generalization gap. Finally, another avenue could be generative processes based on continuous-time Markov chains, beyond standard diffusion and score-matching, including, e.g., jump processes and piecewise deterministic dynamics [BSDB+24, Bau25, BSS+24]. We hope that such extensions will motivate the development of new theoretical tools tailored to broader generative modeling paradigms.

