# OpenReview forum: "Algorithm- and Data-Dependent Generalization Bounds for Diffusion Models"
_NeurIPS.cc/2025/Conference — NeurIPS 2025 poster_

### Official Review · Reviewer_TFd7 · 2025-06-29

**Clarity:** 3
**Significance:** 2
**Originality:** 3
**Rating:** 5
**Confidence:** 3

**Summary:**

This paper analyzes both the sample complexity (data-dependent) and the algorithmic generalization ability (algorithm-dependent) of the score-matching error that arises when training score-based generative models using practical optimization algorithms. The theoretical analysis investigates the generalization behavior of using SGLD and ADAM to solve the empirical score-matching objective, offering insights into how hyperparameters and topological complexity measures influence the generalization error in the context of training score-based diffusion models.

**Questions:**

**Q1:**
In Theorem 4.1, the loss function $\ell_{\lambda}(w, Z)$ is assumed to be $\tau^2$-subgaussian. Is this assumption directly implied by Assumptions 2.1 and 3.1? Similarly, Theorem 4.2 assumes that $\ell_{\lambda}(\theta, z)$ is uniformly bounded with respect to both $\theta$ and $z$. Does this automatically follow from Assumptions 2.1 and 3.1?

---

**Q2:**
How should we interpret the three topological complexity measures introduced in Section 4.3 in the context of score-based diffusion models? Do these quantities provide practical guidance for model training?

---

**Q3:**
Figure 1 suggests that using a large batch size makes training less sensitive to the choice of learning rate and also improves generation quality. Can the theoretical results offer any insight into this empirical observation?

---

**Q4:**
Beyond the choice of training hyperparameters, the noise scheduling in the forward process [1] is known to significantly affect the performance of score-based generative models. Is it possible to incorporate the scheduling of the forward process into the current theoretical framework to offer guidance on how to choose or optimize the noise schedule?

---

**Q5:**
The results in Theorems 4.1 and 4.2 appear to be independent of model complexity. Could the authors comment on why this is the case? Is it correct to interpret that model complexity only influences the empirical explicit score-matching loss (i.e., the first term in the error decomposition in Theorem 3.1)?

---

**Q6:**
Could the authors provide additional numerical results illustrating how the empirical explicit score-matching loss depends on the hyperparameters and the other complexity measures introduced in the paper? For instance, similar to Figures 2 and 3, but tracking the score-matching loss instead of the generalization gap.

[1] Karras, Tero, et al. "Elucidating the design space of diffusion-based generative models." Advances in neural information processing systems 35 (2022): 26565-26577.

**Ethical Concerns:**

["NO or VERY MINOR ethics concerns only"]

**Final Justification:**

I have read all the reviewers’ comments and the corresponding replies from the authors (including my own). In my view, most of the points I raised have been satisfactorily addressed. I find this paper to be theoretically well‑grounded, and I believe the proposed decomposition of the score‑matching error will be of interest to the community. Accordingly, I recommend acceptance.

**Limitations:**

Yes

**Quality:**

3

**Strengths And Weaknesses:**

**Strengths:**

1. The paper is clearly written and provides a helpful overview of prior work, effectively situating its contributions within the existing literature.

2. The theoretical results are well presented and, to the best of my knowledge, offer novel results in the context of score-based diffusion models.

---

**Weaknesses:**

1. Based on my understanding, Theorems 4.1 and 4.2 are adapted from existing results. However, the required assumptions are not explicitly verified or discussed in sufficient detail.

2. The topological complexity quantities introduced in Section 4.3 are not intuitive, and the paper does not provide adequate interpretation of these terms within the context of diffusion models. It would also be helpful to have some discussions on how the result differs from standard settings where ADAM is used to minimize generic training objectives.

3. The theoretical findings offer limited insight into the empirical success of score-based diffusion models or practical guidance for their training.

Please see the questions related to above three points in the Question section.

---

> ### Author Rebuttal · Authors · 2025-07-30
>
> We thank the reviewer for their insightful comments. In the following, we address all the remarks and questions.
>
> **Weakness 1** and **Q1**
>
> We emphasize that Theorems 4.1 and 4.2 are not only consequences of existing results, but also from our theory, developed in Section 3. Indeed, the conclusion of Section 3 is that we boiled down the problem of score approximation to a well-defined statistical learning problem (the generalization error of Theorem 3.1), where a loss function $\ell$ is well-defined. Then, our theorems should be thought as a continuation of Section 3 by exploiting this setting to involve the benefits of statistical learning (i.e. the dynamics of the considered optimization algorithm) for score-based diffusion models.
>
> We thank the reviewer for the question of satisfying the assumptions of Theorems 4.1 and 4.2. Let us first note that the sub-Gaussian or bounded loss assumptions are very classical in the learning theory literature [HNK+20,MWZZ18,NHD+19].
> Verifying these assumptions will depend simultaneously on the *dataset* and the *architecture* of the score network, as they refer to the denoising score-matching loss.
> While it might not be a direct consequence of the assumptions required by our theory in Section 3, we expect that these conditions on the denoising score matching loss can be fulfilled in simple settings, like image datasets.
> In the final version of the paper, we will add a discussion on satisfying these assumptions for particular data structures and leave a more precise understanding of this matter to future work.
>
> **Weakness 2** and **Q2**
>
> Thank you for this insightful remark. These topological complexities have been used in several previous works (see [ADS+24] and references therein). They intuitively represent different notions of ``complexity'' of the trajectory of the optimizer (eg, SGD, ADAM, ...) near a local minimum, as is briefly explained in the appendix. They essentially compute the "intrinsic dimension" of the local minimum found by the optimization algorithm (by looking at the topological properties of the trajectory), hence, they are expected to perform better than bounds that are based on the parameter count, which is always larger than the intrinsic dimension.
>
> In the final version of the paper, we will use the additional space to provide a more intuitive explanation of the meaning of this quantity.
>
> Concerning how these quantities should be interpreted within the context of diffusion models, let us detail this.
> Intuitively, the trajectory of the optimizer should be more complex when the learning task is more difficult. In the context of diffusion models, it means that several parameters should affect it like, for instance, the discretization step of the forward process simulation, as the loss is more sensitive for small times.
> In the final version, we will add a discussion about these aspects. Thank you for raising this and help us improving the quality of our work.
>
> **Weakness 3**
>
> One of our goal is to relate the performance of diffusion models to the optimization dynamics of the score network. For instance, this allows us to connect (part of) the score estimation error to the norms of the gradients encountered during training. To the best of our knowledge, such theoretical observations are new in the context of diffusion models.
> Therefore, we believe that this provides insights towards better training of diffusion models. For instance, Theorem 4.1 is supporting the idea that the generalization error of diffusion models should improve when the model converges to a flat minimum in the parameter space, as it is commonly found in the supervised learning literature [1,2].
>
> **Q3**
>
> The batch size (as well as some other hyperparameters) does not appear explicitly in our theoretical contributions.
> However, note that these hyperparameters appear implicitly through the complexity measures that we use in Theorems 4.1 and 4.2 (through the gradient norms and the topological complexities). Figure 1 reports the Wasserstein metric and the FID calculated on the test set.
> We see in Figure 1 that both metrics decrease, stabilize and then increase for higher learning rates.
> This illustrates the existence of a trade-off, in the sense that the test error depends on both the optimization error and the generalization error, whose analysis is not the aim of this paper.
> Finally, Figure 1 was obtained using very simple models and datasets and its purpose is to illustrate that the learning algorithm affects the test performance of diffusion models, hence, justifying our approach and our main decomposition.
>
> **Q4**
>
> Incorporating the noise schedule is definitely an interesting suggestion by the reviewer.
>
> In our setting and under our assumptions (especially the finite Fisher information), we believe that it is non trivial to explicitly factor the noise schedule in our contributions and we leave this fr future work.
> However, the reviewer is right to mention that the noise schedule is known to affect the performance.
> In particular, as the noise schedule affects the properties of the loss function, should have an impact on the optimization dynamics and, hence, on the generalization error term (i.e., the second term $G_\lambda$ in our decomposition). Therefore, the effect of the noise schedule is implicitly taken into account by the complexity measures studied in Section 4.
> In the final version, we will add a discussion about this remark from the reviewer.
>
> **Q5**
>
> It has been a long standing topic in learning theory that what is a good notion of "model complexity" for neural networks. It has been shown in numerous works that the good notion is definitely not the number of parameters of the model.
>
> Within this context, the results in Section 4 suggest two notions of model complexity, that are implicitly measured by the quantities appearing in Theorems 4.1 (gradient norm) and 4.2 (topological complexity).
> Moreover, our framework is compatible with more classical measures of the complexity of the model class, such as the Rademacher complexity, so that they could be incorporated in our analysis of the second term of the decomposition (the generalization error). However, we believe that such complexity metric would have little empirical interest but we are open to any suggestion of the reviewer regarding other pertinent complexity metrics.
> To sum up, one of the goals of our paper is to introduce a notion of model complexity through the analysis of the second term on our decomposition. Our theoretical contributions allow us to do that by upper bounding the other term in the decomposition.
>
> **Q6**
>
> We thank the reviewer for the opportunity to further clarify the precise role of the complexity measures introduced in our work and their relationship with the optimization process.
>
> Because of a recent update to Neurips guidelines, we are not allowed to include figures in our rebuttal and have enough characters for only one table. However, we have produced additional tables to satisfy the reviewer's request, tracking the denoising score-matching (DSM) loss against our complexity measures. Indeed, the DSM loss is equal to the empirical explicit score-matching (ESM) loss up to a constant independent of model parameters. We will also include these experiments, along with more figures, in the final version of the paper.
>
> Across these experiments, contrary to the generalization gap, we observe no simple relationship between the train ESM loss and our complexity measures, as clearly exemplified in Table 1, where a low complexity value is observed concurrently with high and low train loss values. In Table 1, where all hyperparameters achieve a similarly low ESM loss, we see non-trivial generalization patterns showing a clear positive correlation with our topological or gradient norms-based complexity measures. This confirms that we have not collected spurious observations from trivial regimes and that our complexity measures are relevant quantities
>
> | Test DSM | Train DSM | Eval Gradient Norms Bs | Eval E Alpha | Eval Positive Magnitude Small | Eval Positive Magnitude Sqrt N | Optim Lr | Training Batch Size |
> |:---:|:---:|:---:|:---:|:---:|:---:|:---:|:---:|
> | 0.0773 | 0.0296 | 0.042 | 480.8 | 27.29 | 4396 | 0.000100 | 4.00 |
> | 0.0811 | 0.0128 | 0.087 | 564.7 | 42.88 | 4605 | 0.000100 | 16.00 |
> | 0.0842 | 0.0262 | 0.082 | 453.8 | 28.18 | 4290 | 0.000050 | 4.00 |
> | 0.0851 | 0.0126 | 0.207 | 558.5 | 41.98 | 4573 | 0.000050 | 16.00 |
> | 0.0859 | 0.0073 | 0.526 | 615.8 | 52.18 | 4698 | 0.000050 | 64.00 |
> | 0.0865 | 0.0074 | 0.242 | 628.1 | 52.91 | 4712 | 0.000100 | 64.00 |
> | 0.0894 | 0.0129 | 0.959 | 591.4 | 41.97 | 4625 | 0.000020 | 16.00 |
> | 0.0948 | 0.0084 | 2.47 | 746.6 | 51.74 | 4835 | 0.000020 | 64.00 |
> | 0.0959 | 0.0066 | 0.397 | 637.3 | 58.81 | 4731 | 0.000100 | 128.0 |
> | 0.0972 | 0.0264 | 0.337 | 460.4 | 27.67 | 4302 | 0.000020 | 4.00 |
> | 0.0998 | 0.0266 | 1.27 | 497.6 | 27.42 | 4391 | 0.000010 | 4.00 |
> | 0.1023 | 0.0065 | 0.932 | 667.2 | 57.14 | 4757 | 0.000050 | 128.0 |
> | 0.1071 | 0.0146 | 3.69 | 683.2 | 40.22 | 4735 | 0.000010 | 16.00 |
> | 0.1116 | 0.0075 | 4.22 | 840.7 | 57.32 | 4894 | 0.000020 | 128.0 |
> | 0.1134 | 0.0094 | 9.89 | 831.0 | 50.90 | 4879 | 0.000010 | 64.00 |
> | 0.1362 | 0.0085 | 19.38 | 1005 | 55.01 | 4942 | 0.000010 | 128.0 |
>
> Table 1: ADAM optimizer on the flowers dataset. Train/test Denoising Score-Matching loss (DSM) vs several complexity metrics and corresponding learning rate and batch size.
>
> [1] Kaiyue Wen, Zhiyuan Li, and Tengyu Ma. Sharpness Minimization Algorithms Do Not Only Minimize Sharpness To Achieve Better Generalization. In Conference on Neural Information Processing Systems, 2023.
>
> [2] Stanislaw Jastrzebski, Zachary Kenton, Devansh Arpit, Nicolas Ballas, Asja Fischer, Yoshua Bengio, and Amos J. Storkey. Three Factors Influencing Minima in SGD. arXiv, abs/1711.04623, 2017.

---

> > ### Comment · Reviewer_TFd7 · 2025-08-03
> > **Reply to the authors**
> >
> > I would like to thank the authors for providing detailed explanations that address my questions and concerns. In my view, most of the points I raised have been satisfactorily addressed, and accordingly, I have raised my score.
> >
> > Regarding the compatibility of Assumptions 2.1 and 3.1 with those required for Theorems 4.1 and 4.2, I understand that it may not be straightforward to directly verify their compatibility. Nonetheless, it would be valuable for the authors to explicitly acknowledge any potential theoretical gap and to discuss how those assumptions are satisfied in practice. I recommend that the authors include a remark in the final version elaborating on this asepect.

---

> > > ### Author Response · Authors · 2025-08-04
> > > **Thank you for your answer**
> > >
> > > Dear Reviewer TFd7,
> > >
> > > We thank you for your very detailed review and your positive appreciation of our rebuttal.
> > > Your remarks help us improve our paper.
> > >
> > > As we mentioned in the rebuttal, the links between Assumption 2.1 and 3.1 and the conditions required in Section 4 is an intricate question.
> > > We will acknowledge this in the final version, as requested. Moreover, we will include the discussion of the rebuttal on this matter and discuss cases where these conditions could be satisfied.
> > >
> > > Thank you again for your positive rating.

---

### Official Review · Reviewer_Bifm · 2025-07-02

**Clarity:** 2
**Significance:** 2
**Originality:** 1
**Rating:** 4
**Confidence:** 3

**Summary:**

This paper delves into the statistical analysis of diffusion models and is the first to take the Dirac sampling from the dataset and optimization generalization error into account. They further conduct experiments to show the effect of optimization dynamics on the generalization ability of the DMs.

**Questions:**

See weakness

**Ethical Concerns:**

["NO or VERY MINOR ethics concerns only"]

**Final Justification:**

As mentioned in the discuss phase, I believe this work benefits both academic and industry in learning the generalization bound of diffusion model.

**Quality:**

3

**Strengths And Weaknesses:**

**Strengths**

1. The mathematical derivation is rigorous.
2. The experiments and figures are illustrative to show the effect of optimization dynamics on the generalization ability of DMs.

**Weaknesses**
1. The generalization analysis of the optimization algorithms like SGLD and ADAM directly follows the previous works, but only adopts them into the diffusion models’ score matching analysis.
2. I am not sure if the ‘Dirac sampling from data distribution’ is both the source of the data-dependent constant and the score generalization gap in your analysis. I think a more heuristic explanation of the source of the different error parts presented in your analysis is needed.

---

> ### Author Rebuttal · Authors · 2025-07-30
>
> We thank the reviewer for their feedback. We believe that we have addressed all the raised issues. We hope that the reviewer could reconsider their score if they believe their concerns are resolved.
>
> **Weakness 1**
> We emphasize that the contributions of Section 4 are more diverse than direct adaptations of existing results, let us develop this below.
>
> The key contribution of our work is showing that these results, which were originally developed in the context of supervised learning, can be meaningfully applied to the analysis of diffusion models through the lens of score matching. This adaptation is enabled by our theoretical reduction (Theorem 3.1), which bridges the gap between generative modeling and the generalization of score functions. While the bounds themselves are not new, their application in the generative setting is, and we believe this is an important and nontrivial contribution. Hence, our results are original within this context.
>
> To be more precise, in Theorem 3.1 (our main decomposition), we unveil the impact of three terms. The main theoretical contribution of this work regards the decomposition itself and our bound on the third term (the data-dependent constant), most of our theoretical efforts are focused on this part. Thus, while Theorems 4.1 and 4.2 rely on existing results, Section 4 brings up original results in two distinct ways.
>
> *Theorems 4.1 and 4.2 draw links between diffusion models and statistical learning.*
> Our theorems should be thought as a continuation of the theoretical results of Section 3. Indeed, the conclusion of Section 3 is that we boiled down the problem of score approximation in diffusion models to a well-defined statistical learning problem (the generalization error of Theorem 3.1), where a loss function $\ell$ is well-defined. Then, the results of Section 4 exploits this setting to involve the benefits of statistical learning (i.e. the dynamics of the considered optimization algorithm) for score-based diffusion models.
>
> *Theorems 4.1 and 4.2 allow novel optimization-based correlation measures.*
> Theorems 4.1 and 4.2 should also be considered as ways to obtain novel and practical correlation measures of the generalization ability of diffusion models. To our knowledge, it is the first time that the benefits of the optimization algorithm are involved as a measure of generalization of diffusion models. We insist that to highlight this strength, we computed practical experiments showing the relevance of these correlation measures on several datasets.
>
> To sum up, while Theorems 4.1 and 4.2 exploit existing generalization bounds, they should be thought simultaneously as the final development of our theoretical contributions (expanded in Section 3) and the first step towards practical experiments, by unveiling original correlation measures that are tailored for diffusion models.
>
>
> **Weakness 2**
>
> As the reviewer mentions, our theory makes use of a finite dataset sampled i.i.d. from the data distribution (hence, a Dirac sampling from the data distribution).
> This is a classical setting in learning theory, as in practice, models are trained on a finite dataset.
> As it can be seen in Theorem 3.1, this empirical distribution of the dataset plays an important role in every term of the proposed decomposition of the score approximation error, as the reviewer suggested.
> Let us make it more clear by providing a more intuitive explanation of these terms.
>
> The first term (empirical explicit score matching loss) corresponds to the quantity that is being minimized by the learning algorithm on the dataset, and it is the lower bound of the empirical denoising score-matching loss.
>
> The third term represents (the data-dependent constant) a statistical bias that only depends on the data distribution and the dataset, i.e., not on the score network.
> Therefore, the existence of this term is a consequence of the Dirac sampling of the data distribution.
> It is a characteristic of the learning problem and providing an upper bound on this term is another main contribution of this work.
>
> Finally, the remaining term (second term) is the generalization error associated with the denoising loss.
> Again, this term is non-trivial because of the presence of the empirical distribution of the dataset in the optimization process.
> This allows us to exploit the rich literature on generalization bounds and to apply, for the first time, classical generalization bounds to diffusion models. Note that we used as our examples existing bounds for SGLD and ADAM, but other generalization bounds would be applicable to our framework too, hence, highlighting its generality.
>
> In the final version of the paper, we will add more details on the interpretation of these terms, as asked  by the reviewer.

---

> > ### Comment · Reviewer_Bifm · 2025-08-05
> > **Thanks for the rebuttal**
> >
> > After going through the detailed rebuttal, I believe most of my concerns are well addressed, especially I found the analysis in Sec. 4 is more diverse than I thought. I believe the study of generalization bound of diffusion model is manfully to both academic and industry thus I am glad to raise my rating accordingly. It is also highly suggested the author can include the discussion in their manuscript, as they confirmed.

---

> > > ### Author Response · Authors · 2025-08-08
> > > **Thank you for your review**
> > >
> > > Thank you for re-evaluating your score. We confirm that we will incorporate all the discussion you raised in the next version of our work to highlight precisely the originality of our contribution.
> > >
> > > We thank you for your time.

---

> ### Comment · Area_Chair_M22b · 2025-08-05
>
> Dear reviewer,
> If not already, please take a look at the authors' rebuttal, and discuss if necessary.
> Thanks,
> -AC

---

### Official Review · Reviewer_x95P · 2025-07-02

**Clarity:** 4
**Significance:** 3
**Originality:** 3
**Rating:** 5
**Confidence:** 3

**Summary:**

This paper provides a theoretical bound on the Kullback-Leibler (KL) divergence between the data distribution and the generated distribution of score-based generative models based on Ornstein-Uhlenbeck process.
The paper presents a key decomposition of an approximation error term of the score function, enabling further analysis of generalization performance based on empirical datasets and optimizers.
Using this decomposition, the paper derives a gradient-based generalization bound for stochastic gradient Langevin dynamics (SGLD) and a topological generalization bound for ADAM.
Experiments on image datasets demonstrate a correlation between the generalization error and the derived upper bounds.

**Questions:**

1. Could you please clarify the connection between the presented theoretical results and the limitation (i) mentioned in L73 regarding practical classes of score estimators?
2. Could you please address the errors in Weaknesses 2 and 3?

**Ethical Concerns:**

["NO or VERY MINOR ethics concerns only"]

**Final Justification:**

During the rebuttal, the authors clarified that the presented theoretical results implicitly take into account model architectures, in addition to optimizers and data structures. The authors also stated that the formatting issues of Figures 4 and 5 would be addressed accordingly. Therefore, all issues I pointed out have been adequately addressed.

**Limitations:**

yes

**Paper Formatting Concerns:**

No concerns

**Quality:**

3

**Strengths And Weaknesses:**

### Strengths
The paper is well-structured. The introduction clarifies the target issues of previous work and provides a clear overview of the theoretical results. The logical flow is easy to follow, and the purpose of each section is clearly defined. The limitations of the work are well-discussed.
The paper demonstrates the usefulness of Theorem 3.1 and Proposition 3.1 by extending algorithm and data-dependent generalization bound analyses based on these derived theorems.
The experimental results on image datasets support the presented theoretical claims.

### Weaknesses
1. In the introduction (around L73), the lack of consideration for practical model architectures such as UNet is raised as an issue of an existing approach. However, the algorithm-dependent generalization analysis in the paper focuses primarily on optimizers, and the effects of model architectures do not appear to be discussed.
2. In Figures 4 and 5 in the Appendix, the vertical axes of the bottom right figures appear unusual. They repeat the same value as the tick labels.
3. There are some minor typos and missing information. I have listed some of them below:
    * There appears an unnecessary parenthesis ( within s(T-t, \overleftarrow{X_t}) in (3).
    * The order of arguments of s_{\hat{\theta}^{(n)}} in (6) seems to be swapped; it is supposed to be (s, t) instead of (t, s).
    * Just above (6), t\in[t_k, t_{k+1}] is supposed to be the subscript of (\hat{X}_t^{(n)}).
    * The definition of B(0,D) is missing in Assumption 3.1. It likely refers to the ball centered at 0 with radius D.
    * \sigma^2 around L669 does not appear to be introduced, and it should consistently be 2 throughout the paper.
    * There is a word duplication: "have have" in L696.
    * In the equations below L680, d\nu(x) is supposed to be d\nu(z).
    * In L715 and L716, the paper states that Hölder's inequality was used to derive inequalities. However, it seems more likely that Jensen's inequality was used.
    * \zeta around L754 is supposed to be \delta.

---

> ### Author Rebuttal · Authors · 2025-07-30
>
> We thank the reviewer for their detailed review, which will help improve our work.
>
> **Weakness 1 and question 1**, **the algorithm-dependent generalization analysis in the paper focuses primarily on optimizers, and the effects of model architectures do not appear to be discussed.**
>
> Thank you for this insightful remark. Indeed, our bounds (Theorems 4.1 and 4.2) have no explicit dependence on the model architecture, but an implicit one.
> More precisely, our theory does not require explicit assumption on the structure of the model class, which makes it applicable to a broad range of architectures, even the ones that are used in practical applications, such as the UNet.
> The generality of this approach stems from the generalization bounds we used, derived from PAC-Bayes theory (Alquier 2024), which allow considering any architecture by relocating the intrinsic properties of the model within a KL divergence term, translated here by the gradient norms (Thm 4.1) and the topological complexity measures (Thm 4.2). These terms take implicitly into account the model architecture in our generalization bounds, as it is clear that the model architecture will have an impact on the gradient norms, for instance.
> In the next version of the paper, we will add a precise discussion about this matter.
>
> **Weakness 2**
>
> Thank you for catching this issue regarding the axis of some figures in the appendix, we will correct it in the final version of the paper. This formatting issue is due to the fact that in these experiments, the positive magnitude with scale $\sqrt{n}$ saturates to its maximum value, so all points on the y axis have approximately the same value. The fact that the positive magnitude can saturate when the scale parameter is too high has been documented in previous works, see for instance [ADS+24] and references therein. We reported these bottom-right figures for the sake of completeness; they do not yield significant experimental insights.
>
>
> **Weakness 3** and **Question 2**
>
> We thank the reviewer for pointing out several typos, we will correct all of them in the final version of the paper. Here are a few additional comments.
>
> - The notation $B(0,D)$ indeed refers to the ball of radius $D$, we will add this definition to the assumption.
> - L669 $\sigma$ is indeed a notation for $\sqrt{2}$ it is a typo and will be replaced by $\sqrt{2}$.
> - L715 and L716, by Hölder's inequality we mean that we use the inequality $(a + b)^j \leq 2^{j-1}(a^j + b^j)$, which is also a consequence of Jensen's inequality. We will clarify it.
>
> Reference: Alquier 2024 User-friendly Introduction to PAC-Bayes Bounds

---

> > ### Comment · Reviewer_x95P · 2025-08-05
> >
> > I appreciate the authors for their thoughtful responses.
> > I now understand that the terms implicitly account for the model architecture.
> > I would like to maintain my positive rating.

---

> > > ### Author Response · Authors · 2025-08-08
> > > **Thank you for your review**
> > >
> > > We thank you again for your time and your careful review, which helps to improve the clarity of our work. We will incorporate all the discussions you raised in the next version of our work.

---

> ### Comment · Area_Chair_M22b · 2025-08-05
>
> Dear reviewer,
> If not already, please take a look at the authors' rebuttal, and discuss if necessary.
> Thanks,
> -AC

---

### Official Review · Reviewer_6v73 · 2025-07-02

**Clarity:** 4
**Significance:** 3
**Originality:** 2
**Rating:** 5
**Confidence:** 4

**Summary:**

This paper analyzes in more detail the generalization error of learning a score function. They propose a new decomposition of the score error term that is commonly assumed to be small (under large hypothesis classes) and analyze each term of the decomposition separately, providing concentration bounds and ending with progress towards end-to-end analysis of the error of learning a score. In particular, the score error is broken up into 3 terms, one involving the inherent loss of the score function trained on the data, and the other two representing errors that occur from using an empirical dataset rather than the true distribution. These terms represent the generalization gap. These latter terms are then analyzed and bounded. They also show numerical experiments that support their theoretical claims.

**Questions:**

1. Can assumption 3.1 be weakened to be bounded with high probability? being strictly bounded is a little strong, considering Gaussians have exponentially small (but nonnegative) infinite tails. Perhaps some subgaussian assumption instead?
2. Is Figure 3 just evidence that supports the theorem by [MWZZ]? Is there anything represented by the graphs that is meant to represent theory found in this paper?
3. In general, is there anything in section 4 that is particularly novel to this paper?

**Ethical Concerns:**

["NO or VERY MINOR ethics concerns only"]

**Final Justification:**

My main initial concerns were that of novelty. Coming back to this submission a week later, and thinking more about the merits of the results, I am convinced that the framework presented and justified in the paper is worthy of an accept.

**Limitations:**

yes

**Quality:**

4

**Strengths And Weaknesses:**

Strengths:
The breakdown of the score error into three (somewhat) interpretable terms is great, especially since these terms can be bounded. The more careful analysis of the score matching error term allows one to more clearly understand what is going on with the error term. The writing is also clear and seems well-focused on conveying the intuitive meaning of the work.

Weaknesses:
Although the paper does bring up a new way in which to analyze the score error, the analysis of one of the terms is from previous results, and merely borne out in experiments in this work, and another term is (understandably) left as inherent. The third term is bounded using a (somewhat coarse, as admitted by the authors) concentration inequality, which seems to be the main theoretical contribution of the work. Still, the decomposition is novel in itself and serves as a useful tool for further analysis.

---

> ### Author Rebuttal · Authors · 2025-07-30
>
> We thank the reviewer for their insightful feedback. We address below the raised concerns and questions.
>
> **Question 1**
>
> We thank the reviewer for raising this important point. Indeed, in our work, we assume that the data distribution has compact support since it simplifies several technical derivations.
> As the reviewer suggests, we expect that it is possible to weaken this assumption.
> Indeed, the compact support assumption is mainly used for two reasons in our proof: (i) to apply Hoeffding inequality in the proof of Lemma 3.1 and (ii) to obtain estimates of the moments of the norm of the score function. In both cases, it should be enough, at the cost of additional derivations, to only make assumptions on the growth of the moments of the data distribution, e.g., that it is  sub-Gaussian.
> In the final version of the paper, we will add a discussion about this aspect.
>
>
> **Question 2**
>
> The reviewer is right to notice that Figure 3 support the findings of [MWZZ] and others, **in the context of generative modeling** (notice that these bounds were developed for supervised learning), hence it serves as a sanity check to illustrate that the bounds are still meaningful in the generative modeling context.
>
> We would like to underline that our work is the first, to our knowledge, to provide a sound theoretical framework to exploit these previous results (crafting the generalization error of Theorem 3.1 is not obvious) and exploit them in concrete experiments for diffusion models.
> Beyond this original connection, we believe that Figure 3 is necessary to assess that these generalization bounds have empirical relevance when applied to score-based generative models.
> Indeed, such experiments has not been done in the existing literature.
>
>
> **Question 3** and **Weaknesses**
>
>
> We thank the reviewer for assessing the novelty of our proposed decomposition.
> The key novelty of Section 4 is not in deriving entirely new bounds from scratch, but in showing that existing generalization results—originally developed for supervised learning—can be repurposed to analyze generative models. This repurposing is made possible by the new connection we establish in Section 3, which reduces the analysis of generation algorithms to the score generalization term $G_\lambda$. To our knowledge, this is the first time such a transfer of theoretical tools from supervised learning to generative modeling has been made explicit. We believe this is a substantial contribution, as it opens up a rich and previously untapped body of results for use in the generative setting.
>
>
>
> Thus, while Theorems 4.1 and 4.2 rely on existing results, Section 4 brings up original results in two distinct ways.
>
> *Theorems 4.1 and 4.2 draw links between DMs and statistical learning.*
> Our theorems should be thought as a continuation of the theoretical results of Section 3. Indeed, the conclusion of Section 3 is that we boiled down the problem of generalisation in DMs to a well-defined statistical learning problem (the generalization error of Theorem 3.1), where a loss function $\ell$ is well-defined. Then, the results of Section 4 exploits this setting to involve the benefits of statistical learning (i.e, the dynamics of the considered optimization algorithm) for score-based diffusion models.
>
> *Theorems 4.1 and 4.2 allow novel, optimization-based, correlation measures.*
> Theorems 4.1 and 4.2 should also be considered as ways to obtain novel and practical correlation measures of the generalization ability of diffusion models. To our knowledge, it is the first time that the benefits of the optimization algorithm are involved as a measure of generalization of diffusion models.
> To highlight this strength, we computed practical experiments showing the relevance of these correlation measures on several datasets.
>
> To sum up, while Theorems 4.1 and 4.2 exploit existing generalization bounds, they should be thought simultaneously as the final development of our theoretical contributions (expanded in Section 3) and the first step towards practical experiments, by unveiling original diffusion models-tailored correlation measures.

---

> > ### Comment · Reviewer_6v73 · 2025-08-04
> >
> > I thank the authors for their detailed responses to my questions. After thinking on this some more, I am convinced that the framework of analyzing the score error in a way that can leverage a large body of work is indeed quite novel. Since my main concerns were that of novelty and scope, having read the authors' response to my review as well as the authors' responses to other reviews, I will raise my score.

---

> > > ### Author Response · Authors · 2025-08-08
> > > **Thank you for your review**
> > >
> > > We thank you again for your time and your careful review, which will help to improve the clarity of our work. We will incorporate your remarks to the next version of the document.

---

### Note · Authors · 2025-08-14

Dear Area Chair, dear reviewers,

We would like to sincerely thank you for the time and effort dedicated to evaluating our work, and for the constructive exchanges during the discussion phase. These conversations have been highly valuable in clarifying key aspects of our results and refining the presentation of our paper. Below is a brief recap of the discussion outcomes:

-**Reviewer 6v73** engaged with us on the novelty and scope of our Section 4 results. Following our exchanges, they expressed that “the framework of analyzing the score error in a way that can leverage a large body of work is indeed quite novel” and updated their evaluation accordingly.

-**Reviewer x95P** provided technical feedback on the impact of model architecture, identified several typos, and asked clarifying questions about our experiments. Their overall positive view of the work was maintained after discussion.

-**Reviewer Bifm** contributed suggestions regarding the breadth of our Section 4 results, the dataset description, and further clarifications for Theorem 3.1. These points were discussed in detail, and the reviewer acknowledged the diversity of our results in Section 4, updating their evaluation.

-**Reviewer TFd7** raised thoughtful questions on the interpretation of our complexity terms, the scope of Section 4, and our experiments. We addressed these points with detailed explanations and additional experimental results, after which the reviewer confirmed their concerns had been resolved and updated their evaluation.

We greatly appreciate the opportunity to have had these fruitful discussions and **will incorporate all feedback into the next version of our work**.

Authors

---

### Decision · Program_Chairs · 2025-09-17

**Decision:**

Accept (poster)

**Comment:**

This paper considers an quantitative bound of the generalization error of score learning in diffusion model. The key contribution is a refined analysis that is algorithmic- and data-dependent, which is tighter than traditional results that are largely approximation theory based and therefore naturally generic and loose. While training aspect is touched upon, which is a highlight, this paper however overlooked existing works on quantitative bounds of score training dynamics, such as [Evaluating the Design Space of Diffusion-Based Generative Models] and [A precise asymptotic analysis of learning diffusion models: theory and insights]. Along the same line, most reviewers would like to see a clearer clarification on what parts of the paper are based on existing results and what are truly innovative. Nevertheless, all of us agree that the merits of this work outweigh its imperfection, and I'm happy to recommend acceptance. Meanwhile, I strongly recommend the authors to take the reviewers and AC's comments into consideration while preparing for a revised/final version.